# ENTER THE VOID: EXPLORING WITH HIGH ENTROPY PLANS

## ABSTRACT

Model-based reinforcement learning (MBRL) offers an intuitive way to increase the sample efficiency of model-free RL methods by simultaneously training a world model that learns to predict the future. These models constitute the large majority of training compute and time and they are subsequently used to train actors entirely in simulation, but once this is done they are quickly discarded. We show in this work that utilising these models at inference time can significantly boost sample efficiency. We propose a novel approach that anticipates and actively seeks out informative states using the world model's short-horizon latent predictions, offering a principled alternative to traditional curiosity-driven methods that chase outdated estimates of high uncertainty states. While many model predictive control (MPC) based methods offer similar alternatives, they typically lack commitment, synthesising multiple multi-step plans at every step. To mitigate this, we present a hierarchical planner that dynamically decides when to replan, planning horizon length, and the commitment to searching entropy. While our method can theoretically be applied to any model that trains its own actors with solely model generated data, we have applied it to Dreamer to illustrate the concept. Our method finishes MiniWorld's procedurally generated mazes 50% faster than base Dreamer at convergence and in only 60% of the environment steps that base Dreamer's policy needs; it displays reasoned exploratory behaviour in Crafter, achieves the same reward as base Dreamer in a third of the steps; planning tends to improve sample efficieny on DeepMind Control tasks.

## 1 INTRODUCTION

In recent years, reinforcement learning (RL) has achieved remarkable success across a variety of domains, from mastering Go (Silver et al., 2017b) to racing drones at high speed (Kaufmann et al., 2023). However, these successes often rely on dense reward signals and highly structured environments. In real-world applications such as autonomous navigation, exploration, and disaster response, rewards are sparse and environments are stochastic and partially observable. In these conditions, achieving efficient exploration and good sample efficiency remain an active research problem.

Curiosity-based methods typically treat novelty as an intrinsic reward bonus that is added to the environment reward and propagated through the same value- and policy-learning mechanisms as extrinsic reward (Pathak et al., 2017; Burda et al., 2018; Bellemare et al., 2016; Ostrovski et al., 2017; Badia et al., 2020; Raileanu & Rocktäschel, 2020). These mechanisms are derived under a stationary MDP assumption in which the reward function $r(s, a)$ is fixed over time (Sutton & Barto, 2018), so once a bonus has been associated with a state(-action) distribution it is effectively treated as part of a persistent "true" value. In practice, however, novelty-based bonuses are deliberately non-stationary: prediction errors, pseudo-counts, and related signals decay as states are revisited (Ostrovski et al., 2017; Bellemare et al., 2016; Mahankali et al., 2022), and it is common to normalise intrinsic rewards online to stabilise learning under this drift (Burda et al., 2019; 2018). This yields a robust low-frequency signal for long-term novelty seeking, but it remains retrospective: the bonus only reflects novelty after it has propagated through the value function. By contrast, anticipatory exploration methods based on short-horizon predictions of epistemic uncertainty explicitly plan toward states that are *predicted* to be novel before they are experienced (Shyam et al., 2019; Sekar et al., 2020; Jarrett et al., 2023); our approach falls into this anticipatory category.

Instead of having to learn novelty, we instead propose to use model uncertainty. To do this we use model-based reinforcement learning (MBRL), where models of the environment (world models (Ha & Schmidhuber, 2018)) are concurrently trained alongside the actor to predict environment transitions. Usually, the model is used to train the actor and is not used at inference time. We use the model as a planner at inference time to quantify the distribution entropy of the model's predicted next state.

Seeking the next most uncertain state can lead to greedily optimising for noise, while falling into local aleatoric optima, so we use a planner to choose from a series of reward motivated trajectories holistically. To do this, we leverage the world model by combining it with the greedy actor to generate a selection of high scoring rollouts, from which we choose the rollout whose predicted states have the highest entropy and predicted reward. To keep the method reactive, we introduce a meta planner that can terminate planned trajectories when plans become stale and generate a new plan.

Thus, we propose to augment Dreamer (Hafner et al., 2020), a prominent and efficient world model, with a planning mechanism to anticipate and seek informative states as they are about to occur to drive reasoned exploration. This can be applied to any model based reinforcement learning method that trains the actor exclusively with the model (instead of the environment). This method can also be applied in conjunction with curiosity based methods to obtain both long and short term novelty, though we do not focus on that in this work. In this work, we also introduce a lightweight Proximal Policy Optimisation (PPO) Schulman et al. (2017) based hierarchical planner that dynamically decides when to commit to a pre-selected rollout and when to discard it to replan. Importantly, we do not modify Dreamer's world-model or actor training objectives; all losses and KL coefficients remain identical to DreamerV3. Our contribution is an inference-time entropy-seeking planner that changes action selection and thus the replay distribution.

Our contributions are:

- Use an MBRL model as a planner at inference time to proactively target high-entropy states multiple steps (as far as the maximum world model horizon - up to 15 in this work) in the future, supporting seeking delayed gratification via reasoned anticipatory exploration.

- Reinterpret the KL objective in world-model training as a min–max interaction that couples model learning with entropy-seeking exploration, improving information gain and sample efficiency.

- Introducing a reactive hierarchical planner that dynamically selects between committing to a plan and replanning based on new information received, reducing dithering and improving efficiency through learned plan commitment.

The rest of the paper is structured as follows. In Section 2, we review related work in intrinsic motivation, planning, and hierarchical RL. In Section 3, we provide background on Dreamer and its training formulation. Section 4 introduces our entropy-seeking planner and reactive hierarchical policy. Section 5 details our experimental setup, evaluates performance on MiniWorld's procedurally generated 3D maze environment, Crafter, and DMC's vision based control environment. Section 6 concludes this work and outlines limitations.

## 2 RELATED WORK

### 2.1 INTRINSIC REWARD

Intrinsic motivation methods can be grouped into *retrospective* and *anticipatory* approaches. Retrospective methods assign reward after experience has occurred, using prediction error (?), state novelty (Burda et al., 2018), episodic novelty (Badia et al., 2020), or representation surprise (?). While simple and broadly compatible with model-free RL, they are vulnerable to the white-noise problem (attraction to aleatoric uncertainty) and detachment (Burda et al., 2018; Ecoffet et al., 2019). Anticipatory methods instead steer agents toward potentially novel states using short-horizon predictions of epistemic uncertainty (Shyam et al., 2019; Sekar et al., 2020; Chua et al., 2018). It can be argued that retrospective methods look for long term novelty as they attempt to influence the underlying agent behaviour gradually while anticipatory methods often react quickly to emerging novelty, making them a good method to seek short term novelty. We view our method as an anticipatory component that is orthogonal to and composable with retrospective / intrinsic reward methods.

## 2.2 PLANNING

Planning in RL ranges from tree search to trajectory optimization. Monte Carlo Tree Search (MCTS) has proved effective in games (Coulom, 2006; Silver et al., 2016; 2017a) but good performance in these methods assumes (near) full observability and discrete action spaces. Although recent work (Hubert et al. (2021); Antonoglou et al. (2021)) has extended MCTS to stochastic dynamics and continuous actions, these works rely on models being highly accurate and tractable and are not able to be applied to stochastic, partially observed environments. Path-integral / MPPI-style control samples and reweights trajectories under learned dynamics (Gómez et al., 2014; Williams et al., 2015); TD-MPC and TD-MPC2 pair this with TD learning for vision-based control (Hansen et al., 2022; 2023), but the planner's actions can drift from the policy network, risking distribution shift and value overestimation. Closer to our setting, Look Before You Leap prefers low-entropy, high-reward states, which can suppress exploration early in training (Wang et al., 2018); MaxEnt-Dreamer maximizes a discounted entropy of the latent state-visitation distribution via an auxiliary density model and uses this only as a training-time regulariser on the actor, whereas our planner uses the world model's own predictive entropy online to re-rank candidate trajectories, allowing it to react immediately to newly emerging high-uncertainty futures while still preferring rewarding ones (Svidchenko & Shpilman, 2021); RAIF optimizes posterior over prior uncertainty but this necessitates evaluating gains retrospectively, blind to emerging novelty (Nguyen et al., 2024).

## 2.3 HIERARCHICAL POLICIES

Hierarchical RL introduces temporal abstraction via high- and low-level controllers. Option-Critic learns options and termination conditions end-to-end (Bacon et al., 2017), while HiPPO runs PPO at two temporal scales (Li et al., 2019). These methods improve long-horizon credit assignment, but fixed intervals or frequent replanning can limit adaptability; excessive termination also reduces effective commitment. Our planner differs by explicitly learning when to replan versus commit, driven by signals computed from imagined rollouts.

## 3 PRELIMINARIES

We consider a partially observable Markov decision process (POMDP) $P = \langle \mathcal{S}, \mathcal{A}, p, r, \mathcal{X}, Z, \gamma, \rho_0 \rangle$, where $\mathcal{S}$ is the state space, $\mathcal{A}$ the action space, $p$ the transition kernel, $r$ the reward function, $\mathcal{X}$ the observation space, $Z$ the observation kernel, $\gamma \in (0, 1)$ the discount factor, and $\rho_0$ the initial-state distribution. The agent observes pixels $x_t \sim Z(\cdot \mid s_t)$ and acts via a policy $\pi(a_t \mid x_{1:t}, a_{1:t-1})$. A full MDP/POMDP formalism and its connection to Dreamer-style world models is given in Appendix B.

Dreamer (Hafner et al., 2019; 2020; 2023) learns a compact latent dynamics model and performs policy optimization entirely in latent space. In this work, we use DreamerV3 as it is the most recent and advanced formulation of the Dreamer series of models. At its core, Dreamer relies on an RSSM that factorizes the environment into deterministic recurrent states and stochastic latent representations. The RSSM is described by the following formulations:

$$
\begin{aligned}
\text{Recurrent model:} \quad & h_t = f_\phi(h_{t-1}, z_{t-1}, a_{t-1}) \\
\text{Transition predictor (prior):} \quad & \hat{z}_t \sim p_\phi(\cdot \mid h_t) \\
\text{Representation model (posterior):} \quad & z_t \sim q_\phi(\cdot \mid h_t, x_t) \\
\text{Predictors (Image, Reward, Discount):} \quad & \hat{x}_t, \hat{r}_t, \hat{\gamma}_t \sim p_\phi(\cdot \mid h_t, z_t)
\end{aligned}
$$

Here, $x_t$ is the observation at time $t$, and $h_t$ is the deterministic recurrent state. At each step, Dreamer generates a prior latent state $\hat{z}_t$ from the deterministic recurrent state $h_t$ via $p_\phi(\cdot \mid h_t)$, and updates it into a posterior $q_\phi(\cdot \mid h_t, x_t)$ once the new observation $x_t$ has been received. The KL divergence between the prior and posterior is minimized to train the model:

$$
\mathcal{L}_{\text{KL}} = D_{\text{KL}}\Big(q_\phi(z_t \mid h_t, x_t) \,\|\, p_\phi(\cdot \mid h_t)\Big). \tag{1}
$$

Dreamer's policy network is trained using imagined trajectories generated by the world model. This ensures that policy training remains effectively on-policy. The buffer that the world model trains from is populated by a naive $\epsilon$-greedy actor.

# 4 METHOD

## 4.1 ENTROPY

We now connect Dreamer's KL term to an information-gain objective and motivate entropy as a practical, model-aligned uncertainty signal for planning. DreamerV3 models latent states as factorised discrete variables. The latent state $z_t$ is modelled by DreamerV3's discrete RSSM, but the option to use a continuous latent space is also available. Training the model involves minimizing a KL divergence loss between the prior and posterior distributions of the latents (equation 1); classically, the same KL divergence can be viewed as an information gain (IG) term: for a target variable $Y$ and an input $X$, Quinlan's information gain can be written as the reduction in entropy of $Y$ after observing $X$,

$$\text{IG}(Y; X) \;=\; H(Y) - H(Y \mid X) \;=\; \sum_x p(x)\, D_{\text{KL}}\big(p(Y \mid X = x) \,\|\, p(Y)\big) \tag{2}$$

highlighting that information gain is a form of mutual information, expressible as an expected KL divergence (Quinlan, 1986). In our setting, $Y$ corresponds to the latent state $z_t$ and $X$ to the new observation $x_t$. The training objective $D_{\text{KL}}\big(q_\phi(z_t \mid h_t, x_t) \,\|\, p_\phi(z_t \mid h_t)\big)$ is therefore the information gained about $z_t$ from incorporating $x_t$, up to averaging over time. At planning time, however, future observations $x_t$ are not yet available, so the posterior $q_\phi(z_t \mid h_t, x_t)$ cannot be evaluated for candidate action sequences. Any anticipatory uncertainty signal must therefore be computable from the prior alone.

A simple proxy for future information gain is to prefer states whose prior $p_\phi(z_t \mid h_t)$ has high entropy. Intuitively, if the prior over $z_t$ is already low entropy, then, on average over possible observations, little uncertainty can be removed; conversely, a high entropy prior admits the possibility of large reductions in uncertainty. This motivates using the prior entropy as an intrinsic objective for the meta-planner. Thus, our objective becomes maximising prior entropy:

$$\mathcal{J}_t \;=\; \max\; H\big(p_\phi(z_t \mid h_t)\big) \tag{3}$$

The standard entropy functional $H[p] = \mathbb{E}_{X \sim p}[-\log p(X)]$ admits parallel definitions for discrete (Shannon) and continuous random variables, obtained by replacing the sum with an integral (Cover & Thomas, 2006; Shannon, 1948; Marsh, 2013). An advantage therefore of using entropy as an uncertainty measure is that it applies in a unified way across both categorical and Gaussian latent spaces without requiring changes to the model parameterisation.

Thus, by selecting states with high prior entropy, the planner preferentially visits regions where the model's beliefs are uncertain and observations are expected to concentrate the posterior distribution, increasing the KL divergence between the two. The planning objective therefore heuristically maximises the expected KL, while world-model training simultaneously minimises the same KL term, yielding a natural (albeit loose) minmax interaction between exploration and model fitting.

There are two primary failure modes for this uncertainty-based exploration. The first arises in environments with high aleatoric uncertainty, where a state has multiple plausible successors due to inherent stochasticity. In such cases, even a well-understood state $s_t$ may give rise to a high-entropy predictive latent distribution simply because there are multiple legitimate outcomes. This occurs because the RSSM uses a unimodal prior family. During training, the prior $p_\phi(z_{t+1} \mid h_t)$ must match the posterior via the KL term, but the posterior only captures the realised outcome, not the full set of possibilities. When the true next-latent distribution is effectively multi-modal,

$$p^*(z_{t+1} \mid h_t) = \sum_{i=1}^{M} w_i\, p_i(z_{t+1} \mid h_t), \qquad w_i \geq 0, \; \sum_{i=1}^{M} w_i = 1 \tag{4}$$

the RSSM is forced to approximate all modes with a single member of its unimodal family,

$$\hat{p}_\phi(z_{t+1} \mid h_t) \in \mathcal{F}_{\text{uni}}, \qquad H\big(\hat{p}_\phi(z_{t+1} \mid h_t)\big) \;\gg\; H\big(p_i(z_{t+1} \mid h_t)\big) \;\text{ for many } i \tag{5}$$

so that the learned prior entropy is artificially inflated. Here $i$ indexes the distinct plausible successor states and the weights $w_i$ describe their relative frequencies. These states are not inherently bad to be in: they are often bottleneck states that lead to many other states (Ecoffet et al., 2019), some

rewarding and some not. Our agent is naturally encouraged to visit such bottlenecks, but because imagined trajectories are scored by both environment reward and entropy, the greedy actor will favour branches that reach reward-relevant regions, which mitigates pathological fixation on such states.

The second failure mode arises in environments with latent transitions that require specific, rarely executed actions. In such cases, the ideal transition distribution may again be written as a mixture,

$$p^*(\mathbf{z}_{t+1} \mid h_t) = \sum_{i=1}^{M} w_i \, p_i(\mathbf{z}_{t+1} \mid h_t), \qquad w_i \geq 0, \ \ \sum_{i=1}^{M} w_i = 1 \tag{6}$$

but now the weight associated with the common transitions, denoted $w_C$, satisfies $w_C \gg w_{\neg C}$, where $w_{\neg C}$ is the total weight of all rare transitions. If the agent has only encountered the high-probability modes, the learned model will be ignorant of the rare outcomes and will instead estimate a prior dominated by the common component,

$$\hat{p}_\phi(\mathbf{z}_{t+1} \mid h_t) \approx p_C(\mathbf{z}_{t+1} \mid h_t) \tag{7}$$

so that

$$H\big(\hat{p}_\phi(\mathbf{z}_{t+1} \mid h_t)\big) \ \approx \ H\big(p_C(\mathbf{z}_{t+1} \mid h_t)\big) \ < \ H\big(p^*(\mathbf{z}_{t+1} \mid h_t)\big) \tag{8}$$

In these cases, a state's uncertainty may be chronically underestimated and subsequently underexplored. Addressing this likely requires mode-seeking mechanisms (option discovery, social learning, teacher-student learning). While this is outside the scope of the present method, it is possible to amplify this method with future work.

### 4.2 REACTIVE HIERARCHICAL PLANNER

At each environment step we generate $N$ short-horizon imagined rollouts (we use $N{=}256$) starting from the current latent state using the greedy actor and the world model. We then select the rollout trajectory $\tau_\star$ whose latent prior has the highest cumulative entropy *and* highest cumulative reward ($\hat{r}_t$ as predicted by dreamer's internal reward model):

$$\tau_\star = \arg \max_{\tau \in \{\tau^{(n)}\}_{n=1}^N} \sum_{t'=t}^{t+H} \Big( \lambda_r \hat{r}_{t'} \ + \ \lambda_H H\big(p_\phi(\mathbf{z}_{t'} \mid h_{t'})\big)\Big). \tag{9}$$

where both $\lambda_r$ and $\lambda_H$ set the relative weighting of entropy and reward and $H$ is the planning horizon: we use 15 for $H$ as it is the length to which dreamer's world model is trained by default. We then execute the selected trajectory's set of actions (a plan) unless the meta-policy decides to cancel the current plan and generate a new one. We find in practice that our method is not overly sensitive to $\lambda_r$ and $\lambda_H$, we do however test this by ablating just the reward and then just the entropy.

We use a meta planner (a light PPO head) to control when to replan. The meta planner outputs a categorical over $p_t \in \{0, 0.25, 0.5, 0.75, 1\}$ which we squashed as $p_t^2$ to decide replanning: draw $u_t \sim \mathcal{U}(0, 1)$ and replan if $u_t < p_t^2$. We find in practice that excessive replanning is a problem that frequently plagues such planners, a finding echoed across the hierarchical RL literature (Klissarov et al., 2017; Chunduru & Precup, 2022; Johnson & Weitzenfeld, 2025) which is why we change the enaction condition from $u_t < p_t$ to $u_t < p_t^2$, discouraging excessive replanning without explicitly punishing replanning or rewarding commitment. Empirically, the learned meta-policy settles into short but non-myopic commitments, with plan lengths of 2–3 steps on average and occasional longer bursts (see Appendix H, Planner Metrics).

Importantly, planning decisions are re-evaluated at every environment step, allowing for flexible replanning without commitment if needed. Pseudocode for our planning algorithm is given in Appendix C. As input to the PPO policy, we provide the encoder embedding; the RSSM feature vector; the current step number normalized by the episode time limit; the greedy action proposed by the actor; the position within the current plan (normalized); a binary in-plan flag; and the "final" RSSM feature that is predicted to be observed if the current plan is followed to the end.

We maintain replay buffers of meta-transitions with fields: PPO observation, PPO action, PPO sample log prob, implemented flag (whether a replan signal was sent), per-step entropy, next base reward,

and done. The PPO meta-planner is trained on sequence-level returns, aggregating the reward and entropy terms over a planning horizon of length $L$:

$$r_t^{\text{meta}} \;=\; \frac{L}{2} \sum_{t'=t}^{t+L} \Big( r_{t'}^{\text{base}} \;+\; H\big(p_\phi(\mathbf{z}_{t'} \mid h_{t'})\big)\Big). \tag{10}$$

We then compute advantages with $\text{GAE}(\gamma, \lambda)$ and optimize a clipped PPO objective (separate actor/critic) with a naive entropy bonus, using Adam for both policy and value heads. The PPO head trains on all collected transitions. To encourage early behavioral diversity we use He initialization and bias the PPO head's initial logits toward intermediate $p_t$ values (0.25, 0.5, 0.75).

In general, the choice of $L$ controls the temporal scope of the meta-planner's objective: long, sparse-reward or highly delayed-reward tasks benefit from larger $L$, which allows the planner to align its decisions with far-reaching consequences, whereas short-horizon, locally reactive tasks favour smaller $L$ to enable fast adaptation. In our experiments, we set $L=32$ for complex, long-horizon environments such as Crafter, $L=8$ for navigation tasks (MiniWorld Maze), and $L=2$ for short-horizon continuous control tasks (DMC-Vision). This choice lets the reward calculation match the intended commitment level required by each environment.

## 5 EXPERIMENTS

We evaluate across three regimes that stress different aspects of decision making: procedurally generated 3D mazes in MiniWorld Chevalier-Boisvert et al. (2023) for long-horizon navigation under partial observability and sparse reward, Crafter (Hafner (2021)) for survival-style open-world play with diverse subgoals, and vision-based control tasks from the DeepMind Control suite (Tassa et al. (2018)) for closed-loop continuous control. We report means with variability over 5 seeds for MiniWorld [0, 409, 412, 643, 996] and DMC [0, 413, 604, 765, 891], and 5 seeds for Crafter [0, 11, 413, 891, 920]. Because MiniWorld and Crafter are procedurally generated, training and test distributions coincide; MuJoCo-based DMC tasks have no train/test split under our setup; we therefore report training curves only.

We compare with Plan2Explore (Sekar et al. (2020)) as a baseline; it supplies an anticipatory exploration baseline that scores novelty via ensemble disagreement, giving a contrast between disagreement-based uncertainty and our inference-time entropy signal from a single world model. We also add PPO on MiniWorld as a model-free reference (Schulman et al., 2017). For Crafter and DMC, competitive pixel-based model-free methods (augmented SAC/DrQ-style agents) are known to be sensitive to implementation and tuning (Engstrom et al., 2020; Henderson et al., 2018); to avoid confounding factors and keep compute comparable, we omit them here (Kostrikov et al., 2020; Yarats et al., 2021; Laskin et al., 2020; Srinivas et al., 2020). Dreamer is a widely adopted and strong pixel-based MBRL agent that already surpasses earlier model-based and many model-free methods in visual control (Hafner et al., 2019; 2020; 2023).

We train MiniWorld mazes for 350,000 environment steps (approximate convergence for Dreamer under our setup). On DMC and Crafter we use a fixed 24-hour wall-clock budget per method to ensure compute parity. Metrics are task-appropriate: episode time to completion for MiniWorld (shorter is better) and undiscounted training return for DMC and Crafter. All curves use a rolling mean (window 10%) with shaded $\pm 1$ standard error of the mean (SEM) across seeds.

### 5.1 MAZE EXPLORER

We use a 3D maze environment adapted from MiniWorld (Chevalier-Boisvert et al., 2023), where each episode presents a new random layout. The agent receives RGB image observations and performs continuous actions to locate three goal boxes. We introduce a *porosity* parameter that controls wall density to vary exploration difficulty. Observations are augmented with a binary spatial map that encodes visited regions and current orientation, providing a simple episodic memory. The reward function combines exploration, proximity, and goal rewards to encourage both coverage and task success. Full environment details are in Appendix D.

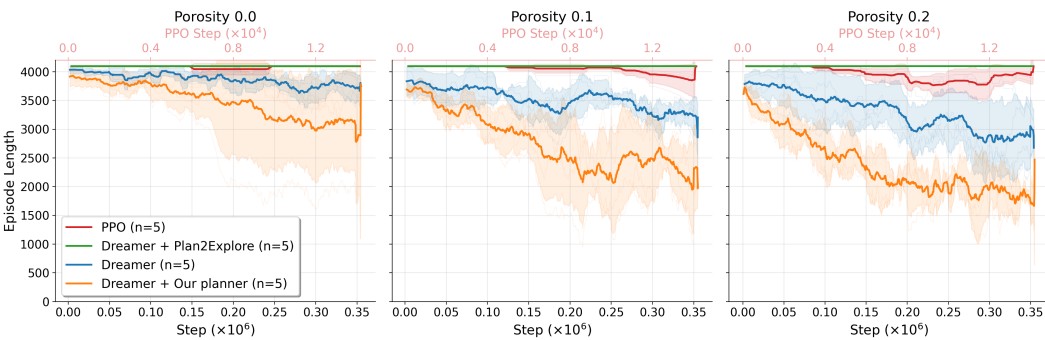

Figure 1: Episode lengths across different porosity levels. Lower porosity increases maze difficulty.

#### 5.1.1 TASK DIFFICULTY

Varying the porosity parameter controls maze difficulty (low porosity leads to difficult mazes). Visual examples of mazes at different porosity levels are provided in Appendix E. Figure 1 shows that our method maintains low episode lengths even in denser mazes, outperforming both Dreamer and PPO. PPO underperforms across all settings, likely due to its lack of memory and long-horizon reasoning. Base Dreamer's performance degrades under low porosity, where rewards are harder to reach, while our method maintains faster episode finishes across porosities in our runs.

Under the most difficult condition (porosity = 0), where only a single path exists between the agent and three goals, both our method and Dreamer perform worse, as shown in Fig. 1. However, our approach still outperforms Dreamer, achieving 20% shorter episode lengths on average, albeit with higher variance due to the increased exploration burden. Plan2Explore underperforms here, which we hypothesize stems from frequent replanning without commitment; ensemble disagreement identifies novelty but does not enforce trajectory-level persistence.

### 5.2 VISION-BASED CONTROL (DMC-VISION)

We evaluate six pixel-control tasks from the DeepMind Control Suite: `cartpole_swingup`, `walker_walk`, `cheetah_run`, `reacher_hard`, `acrobot_swingup`, and `hopper_hop`. Rather than sweeping the entire benchmark, we select a compact set that spans complementary regimes. Agents observe $64\times64$ RGB frames and act in continuous spaces. For each task we train (i) a base Dreamer agent without planning, (ii) our commit-aware planning variant, and (iii) Plan2Explore, all under identical step budgets (in practice they also take up similar amounts of time, more detail is given in the timing analysis section); curves report a rolling mean (window 10%) with shaded $\pm2$ standard error of the mean (SEM) across seeds. Unless otherwise stated, we use a plan horizon of $H{=}16$, a PPO reward length $L{=}2$, sample $N{=}256$ actor-guided candidate rollouts per decision. To isolate reasoned exploration, we use a single environment instance (no vectorization), since heavy parallelism can introduce "free" random exploration. For the same reason, we omit sparse-reward variants (e.g., `cartpole_swingup_sparse`), where neither base Dreamer nor our planner is expected to reliably discover narrow reward regions within the given budget.

A small sensitivity sweep over candidate count and meta horizon on four of the six tasks (cartpole_swingup, walker_walk, hopper_hop, reacher_hard) indicates that gains are stable across a reasonable range of values (Appendix G). We restrict these sweeps to a representative subset to control compute: acrobot_swingup is another underactuated swing-up control task whose planner behaviour closely mirrors cartpole_swingup, and cheetah_run is a high-speed locomotion task already well covered by the hopper_hop and walker_walk sensitivity profiles, so additional sweeps on these two tasks would be computationally costly without providing qualitatively new insights.

Across three of six tasks, the planning variant achieves higher sample efficiency and final return on average. Gains are largest in contact-rich or higher-variance control (Figures 2a, 2f), where short commitments reduce dithering and stabilize control under pixel noise while collecting informative trajectories. On smoother, lower-variance dynamics (Figures 2e, 2d), improvements are smaller but positive on average. Despite `hopper_hop` often benefiting from increased parallelism or

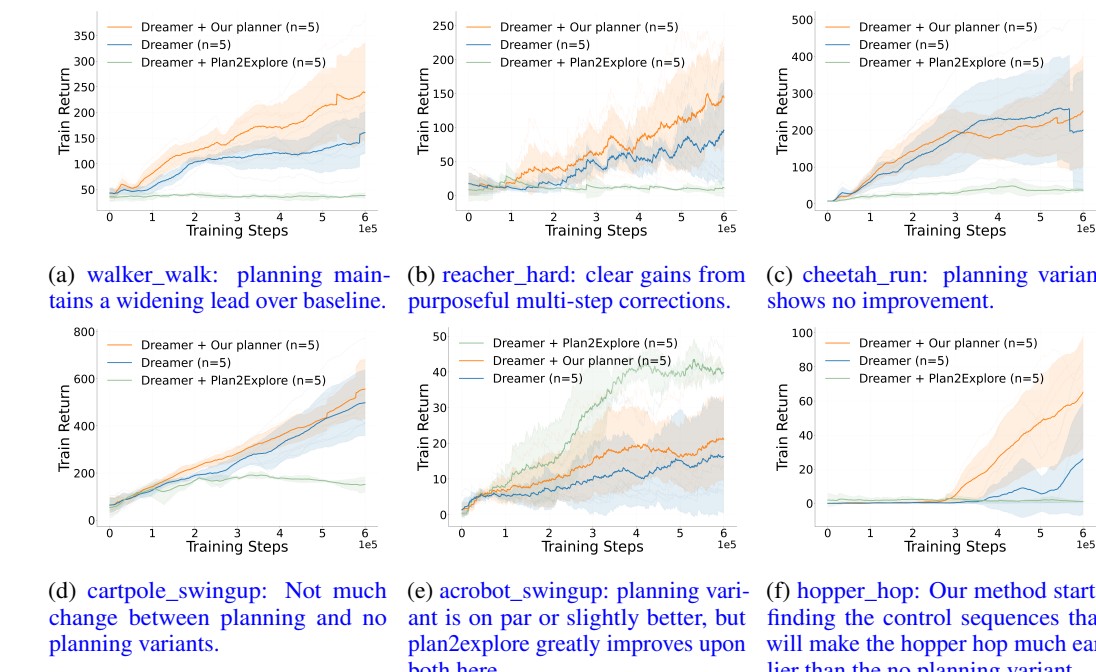

(a) walker_walk: planning maintains a widening lead over baseline.

(b) reacher_hard: clear gains from purposeful multi-step corrections.

(c) cheetah_run: planning variant shows no improvement.

(d) cartpole_swingup: Not much change between planning and no planning variants.

(e) acrobot_swingup: planning variant is on par or slightly better, but plan2explore greatly improves upon both here.

(f) hopper_hop: Our method starts finding the control sequences that will make the hopper hop much earlier than the no planning variant.

Figure 2: DMC-Vision learning curves (return vs. environment steps) for no-plan, the planning variant, and Plan2Explore. Shaded bands: $\pm 2$ standard error of the mean (SEM) across seeds.

longer training, our method discovers effective hopping sequences earlier than no-plan (Figure 2f). Plan2Explore underperforms on most tasks here but is strong on acrobot_swingup, suggesting that disagreement-based novelty aligns with that underactuated swing-up; by contrast, our approach does not detract performance from any task but can make significant improvements on some of the more complex tasks that necessitate sequences of actions.

### 5.3 Open-ended survival (Crafter)

Crafter stresses long-horizon exploration and routine formation. Final episode rewards are given for the number of unique achievements collected, limited to 22. Small penalties and bonuses are given for health changes (Hafner, 2021). We train for 300k environment steps (approximately 24 hours on a GeForce RTX 5090 GPU) and report mean returns with variability over 5 seeds. We run a single environment here as well (no parallel rollouts), which is the default for Crafter. Unless otherwise stated, the planning variant in Crafter is trained with the entropy-only meta objective (and without base reward).

Overall, the planning variant is about 20% higher in average return and reaches comparable thresholds in roughly 50% of the steps that base Dreamer takes (Fig. 3). Because Crafter's reward is dominated by exploratory achievements like crafting tools, placing structures, or engaging enemies, Dreamer's value and policy networks already internalise a form of long-horizon, reward-driven curiosity. Our planner augments this implicit curiosity with an anticipatory entropy-based meta objective, so this comparison can also be read as an ablation between a curiosity-driven agent with and without anticipatory planning. The achievement breakdown in Appendix I shows that the planner-equipped variant acquires repeatable, reward-dense survival routines earlier and more consistently, while Plan2Explore underperforms on Crafter, consistent with the task rewarding sustained multi-step routines rather than one-step novelty chasing.

### 5.4 Ablation Study

To isolate the contributions of individual components, we compare:

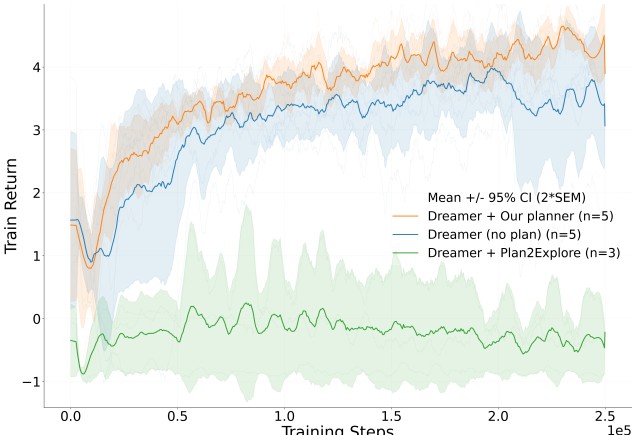

Figure 3: Episode returns during Crafter training.

**Base Dreamer:** The standard agent without planning.

**MPC Style:** To evaluate the value of our underlying planner, we modify the meta planner to replan at every step, mimicking MPC behaviour.

**Meta-reward components:** We ablate the signal used to train the meta-policy by using (i) entropy-only, (ii) reward-only, and (iii) a 50/50 mixture of reward and entropy in $r_{\text{meta}}$.

We run all MPC ablation variants for 90% of the normal experiments' step count to account for the fact that MPC style experiments are slower to run than the full experiments.

Figures 4a and 4b demonstrates the value of committing to plans over extended horizons. Myopic planning causes dithering in both the maze and Crafter environments, with the full planner outperforming MPC-style replanning by leveraging plan commitment. The mixed objective performs slightly worse than either pure objective in the Crafter env, as shown in Fig. 5a, which we attribute to the two terms being only partially aligned in the environment: combining them can reduce selection contrast and dilute the meta-policy's advantage signal, which is why we use pure entropy as our training signal (for Crafter). Importantly, the comparable performance of entropy-only and reward-only suggests that, under actor-guided proposals, reward and predictive uncertainty often point to similar futures. Entropy therefore offers a stable default planning signal when rewards are not available, while remaining competitive when dense rewards are available. The lower variance we observe with entropy is consistent with uncertainty being a more directly model-aligned quantity than predicted reward. In Fig. 8 we see all three planning objectives performing largely the same, except in hopper where entropy-less objectives fail. Hopper has one of the highest reward scarcities in our experimental testbench, without an effective exploration signal it can be hard to perform well in it. Additional sensitivity sweeps over candidate count and PPO horizon are shown in Appendix G.

## 5.5 TIMING ANALYSIS

Our method adds inference-time overhead from generating imagined rollouts and training the meta-policy. With default settings ($H$=16, $N$=256 candidates), a single planning call costs $\approx 0.05$s across all regimes. The cost scales linearly with rollout horizon, while varying the number of candidates has only a small effect due to batched evaluation on GPU. Meta-policy training adds a further $\sim$8-10ms per PPO update at update frequency 32. Because replanning is commit-aware, these costs are amortized over multiple environment steps. Full timing tables and scaling sweeps are reported in Appendix F. We show in Appendix H that the replan rate drops as the agent trains, increasing planner efficiency and possibly reflecting the world model's error rate decreasing with training.

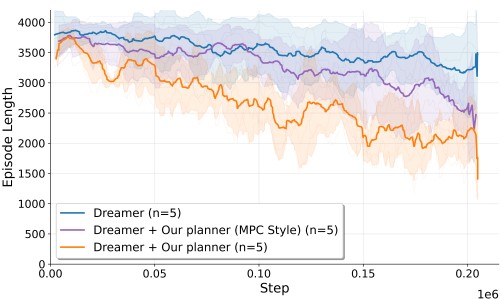 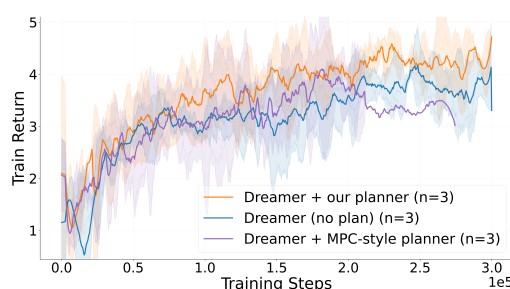

(a) ObjectNav: Episode lengths during training for different ablations. The full planner outperforms both the base Dreamer and MPC-style variants, highlighting the benefit of plan commitment.

(b) Crafter: Episode returns during training for different ablations. The full planner has greater performance and takes less time than the MPC variant, signalling efficiency benefits of plan commitment.

Figure 4: Comparison of episode lengths during training for ObjectNav (left) and Crafter (right) across different ablations.

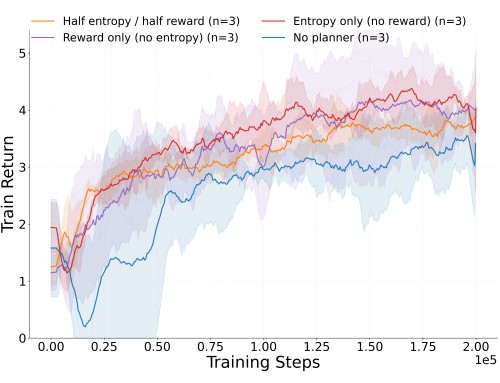 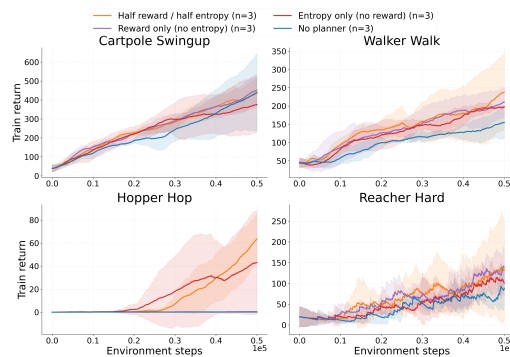

(a) Crafter entropy/reward ablation. All three planner traces outperform the base dreamer variant

(b) DMC entropy/reward ablation. Hopper notably fails without entropy. A larger version of this plot is provided in Figure 8 for clarity.

.

Figure 5: Entropy/reward ablations for Crafter (left) and DMC (right). We compare training the meta-policy with entropy-only, reward-only, and a 50/50 mixture of both. In both domains, entropy-only and reward-only training of the meta-policy are comparable and outperform no-planner, suggesting robustness to the precise meta-reward weighting.

## 6 LIMITATIONS & CONCLUSION

We note that an inherent limitation of our method is that the actor must be trained purely with world model generated states rather than through experience replay; this method biases collection of experiences towards high model entropy, leading to a distributional shift between the actor's policy and the actual behaviour policy. More broadly, we see inference-time entropy planning as a lightweight anticipatory layer for world-model agents. It requires no auxiliary reward learning and fits onto existing objectives. Retrospective intrinsic rewards remain valuable for shaping long-term state distributions, and integrating both signals is a natural direction for future work.

**Reproducibility Statement.** We document datasets, preprocessing, and step-by-step training and evaluation procedures in the Experiments section. Random seeds used for all runs are listed in the experiments section. We will release the complete codebase on GitHub upon acceptance; in the meantime, the paper and appendix provide all details needed to reimplement our results. Configuration settings are located in Appendix J.

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

## A  LLM / AI TOOLING DISCLOSURE

We used AI-assisted tools during this project as follows.

**Tools.**

- **Cursor (AI coding assistant):** Used to generate boilerplate code, suggest refactorings, produce docstrings and unit-test skeletons, and surface API idioms. All produced code was reviewed, modified as needed, and verified by the authors.

- **ChatGPT (writing assistant):** Used for copy-editing, grammar and style suggestions, tightening wording, expanding or condensing paragraphs on request, and clarifying phrasing. We did not use it for ideation, technical contributions, or to generate substantive claims.

The authors reviewed and verified all AI-assisted outputs for correctness and originality, and accept full responsibility for the code and text included in the paper. Any code suggested by tools was tested and adapted to our setting before inclusion.

## B  MDP FORMALISM

We briefly summarise the Markov decision process (MDP) and partially observable MDP (POMDP) formalisms to make the assumptions behind Dreamer and our planner explicit.

**Markov decision process (MDP).**  An MDP is a tuple $\mathcal{M} = \langle \mathcal{S}, \mathcal{A}, p, r, \gamma, \rho_0 \rangle$, where $\mathcal{S}$ is the set of environment states, $\mathcal{A}$ the set of actions, $p(s_{t+1} \mid s_t, a_t)$ the transition dynamics, $r(s_t, a_t)$ the expected immediate reward, $\gamma \in (0, 1)$ the discount factor, and $\rho_0$ the initial-state distribution (Kaelbling et al., 1998). The Markov property, or the memoryless assumption, states that $(s_{t+1}, r_t)$ depend only on $(s_t, a_t)$. A stationary policy $\pi(a \mid s)$ induces trajectories by $s_0 \sim \rho_0$, $a_t \sim \pi(\cdot \mid s_t)$, $s_{t+1} \sim p(\cdot \mid s_t, a_t)$. Following these trajectories, the standard RL objective is to maximise expected discounted return using the Bellman equation:

$$J(\pi) \;=\; \mathbb{E}_{\pi, p}\left[ \sum_{t=0}^{\infty} \gamma^t r(s_t, a_t) \right]. \tag{11}$$

In many environments the agent does not observe $s_t$ directly. A POMDP extends the MDP to $\mathcal{P} = \langle \mathcal{S}, \mathcal{A}, p, r, \mathcal{X}, Z, \gamma, \rho_0 \rangle$, where $\mathcal{X}$ is an observation space and $Z(x_t \mid s_t)$ is the observation (emission) model (Kaelbling et al., 1998). The process evolves as

$$s_0 \sim \rho_0, \tag{12}$$
$$a_t \sim \pi(\cdot \mid x_{1:t}, a_{1:t-1}), \tag{13}$$
$$s_{t+1} \sim p(\cdot \mid s_t, a_t), \tag{14}$$
$$x_{t+1} \sim Z(\cdot \mid s_{t+1}), \tag{15}$$

so the policy must condition on the *history* because $s_t$ is hidden. A sufficient statistic for the history is the belief state $b_t(s) = p(s_t = s \mid x_{1:t}, a_{1:t-1})$, yielding an equivalent fully observed *belief-MDP*. We use the POMDP view throughout, since Dreamer operates from pixels and must infer latent state.

Dreamer's RSSM can be interpreted as learning a compact belief representation for a POMDP: the deterministic recurrent state $h_t$ summarises history, the prior $p_\phi(z_t \mid h_t)$ predicts latent futures, and the posterior $q_\phi(z_t \mid h_t, x_t)$ refines this belief after observing $x_t$, trained via the KL loss in Eq. equation 1. In continual RL, Khetarpal et al. (2022) emphasise that both partial observability and nonstationarity can be modelled by augmenting the hidden state with task/phase variables, i.e., treating continual learning as a (possibly changing) POMDP. This perspective motivates using model uncertainty over latent state as a principled exploration signal: high-entropy priors correspond to broad beliefs about unobserved environment factors, which our planner seeks out in imagined futures.

## C  PLANNING ALGORITHM

---

**Algorithm 1** Entropy Seeking Anticipatory Planning

---

1: **Input:** Observation $o_t$
2: Meta-policy computes discrete planning probability $p_t \in \{0, 0.06, 0.25, 0.56, 1\}$ (via squaring sampled values from $\{0, 0.25, 0.5, 0.75, 1.0\}$)
3: Sample $u_t \sim \mathcal{U}(0, 1)$
4: **if** $u_t < p_t$ **then**
5:     Greedy actor samples $C$ (256) candidate actions $a_1, \ldots, a_C$ from $o_t$
6:     **for** $i = 1$ to $C$ **do**
7:         Roll out trajectory $\tau_i$ of length $H$ (maximum rollout length, 16 here) using world model and greedy actor
8:         Compute $E_H = \frac{1}{H} \sum_{t=1}^{H} E_t^{(i)}$
9:     Select trajectory $\tau_{best} = \arg\max E_H$
10:    Set plan to $\tau_{best}$
11: Continue interacting with environment; repeat planning check at next step

---

## D  MINIWORLD ENVIRONMENT AND REWARD SCHEME

We extend the MiniWorld Maze environment(Chevalier-Boisvert et al., 2023) with several task relevant augmentations. The maze environment is a 3 dimensional procedurally generated maze of size 8x8 (using recursive backtracking) where the agent can take continuous actions along three dimensions - forward/back (step size attenuated if moving backwards to encourage progress), strafe left/right, turn left/right. Each observation consists of a forward-facing RGB image of size (64x64x3). Each episode ends when the time limit (4096) is reached, or the three goal boxes have been found. Each training run samples a new maze structure every episode to prevent memorization. No regions of the maze are sectioned off from the rest of the maze and all the goal states are reachable.

To promote structured exploration, we introduce a porosity parameter that controls wall density: with probability $p$, wall segments are randomly removed during generation. This provides a tunable complexity gradient for navigation tasks by creating variable maze connectivity.

An auxiliary binary 2D map of size (64x64x3) that records agent visitation over the course of an episode has been concatenated to the observation. This map records visited coordinates as 1s whereas unvisited coordinates are kept at 0. The position of the agent and the direction it is looking in is also visible on the map. This serves as episodic spatial memory that enables agents to reason about coverage and connect their actions to the current observation. This mirrors plausible real-world capabilities that can be enacted through GPS tracking or odometry.

The reward function consists of three components:

**Exploration Reward:** A positive reward is granted when the agent visits a previously unvisited cell in its binary exploration map. The reward magnitude is proportional to the number of newly visited cells within a square region around the agent, the size of which is controlled by the *blur* parameter, given by $b$. While this reward introduces non-Markovian dynamics by incorporating visitation history, the inclusion of a binary map in the observation allows memory-less model-free agents such as PPO to perform effectively in this environment.

$$\text{exploration reward} = \begin{cases} \dfrac{\Delta_t}{b^2} & \text{if } b > 1 \\ \Delta_t & \text{otherwise} \end{cases}$$

where $\Delta_t$ = number of newly explored cells at time $t$

**Proximity Reward:** A smoothly decaying signal is emitted by each goal object, with exponentially scaled rewards given when the agent is within an $x$-unit radius. This mimics real-world analogs such as bluetooth signals or radio signals for search and rescue, animal noises for ecological monitoring, or semantic hints for more advanced exploration. This reward

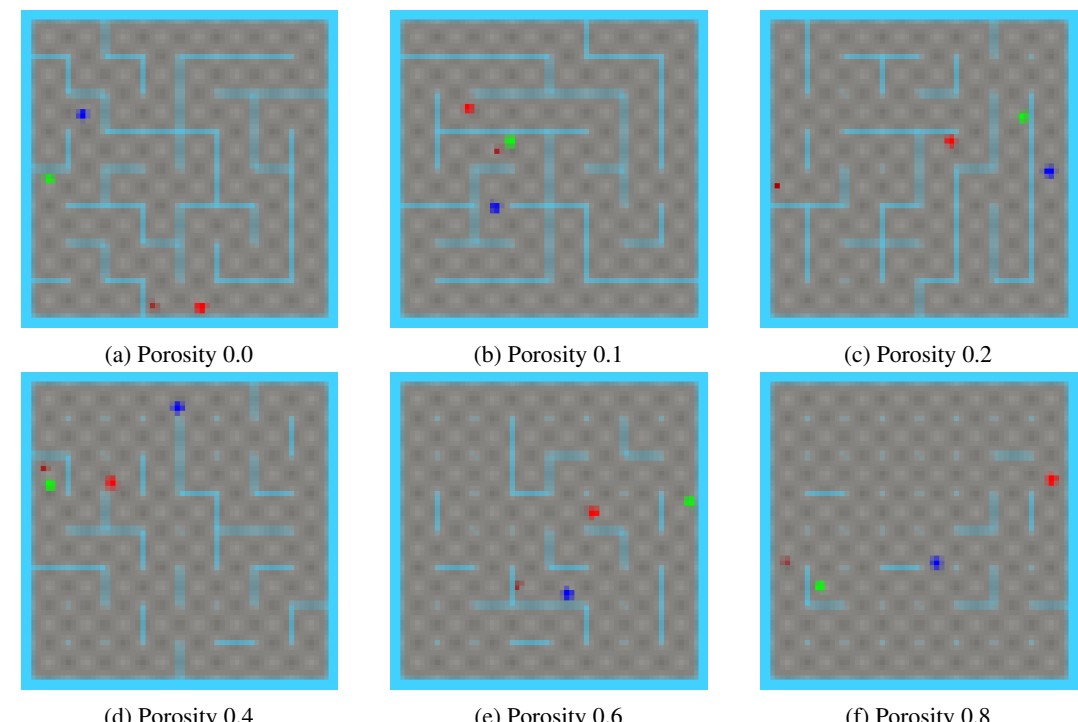

Figure 6: Top-down maze layouts at selected porosity levels. Higher porosity values remove more internal walls, increasing openness and reducing planning difficulty.

takes the form of two bars in the center of the image - if the agent is near a goal box of a particular colour (red, green, or blue), the bars will turn that colour with intensity varying with distance.

$$\text{Proximity reward} = \begin{cases} 0 & \text{if } \Delta < 0 \text{ or } \Delta > 10 \\ (10 - \Delta)^2 \cdot p_{mul} & \text{otherwise} \end{cases}$$

$$\text{where } \Delta = \text{dist} - (r_{\text{agent}} + r_{\text{box}} + s) \text{ and } p_{mul} = 0.03$$

**Goal Reward:** The agent gets a reward for moving into a coloured box. It gets 50 per box and then 150 when it gets the third box.

Thus the overall reward is composed of these three elements summed onto the baseline of -10. The lower limit of reward gained in an episode is $-T$ where $T$ is the time limit, and the upper limit is 0.

## E    MAZE IMAGES

To visualize the effect of varying porosity on maze complexity, we provide top-down views of generated mazes at increasing porosity levels, see Fig. 6. As porosity increases, more internal walls are removed, resulting in more open environments. These top-down maps reflect the structural differences that influence planning difficulty.

To contextualize the agent's perspective within these mazes, we also provide an example of the full map layout and a corresponding visual observation seen by the agent, as given in Figure 7.

## F    TIMING AND COMPUTE OVERHEAD

This appendix reports detailed timing for our planning module and the PPO-based meta-policy. All timings were measured on the same hardware used for the main experiments, with the default planner unless otherwise stated ($H{=}16$, $N{=}256$ candidates).

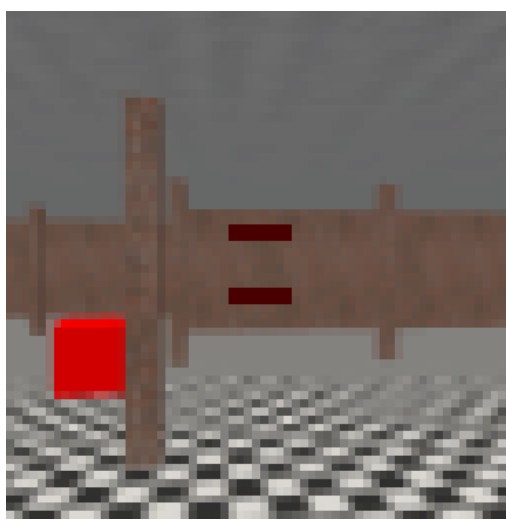
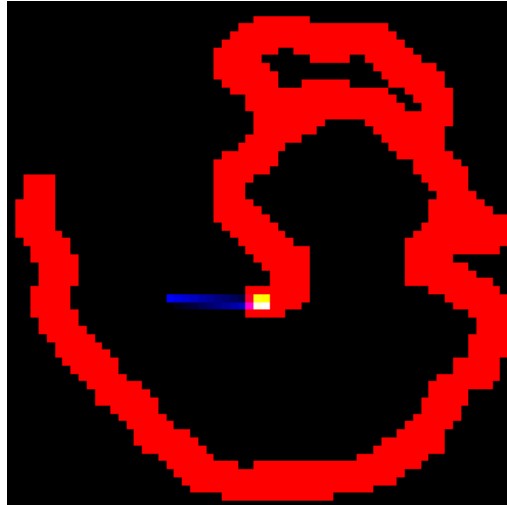

(a) Agent's visual observation       (b) Agent's map observation

Figure 7: Image of what the agent perceives - the visual observation (left) and of the map observation (right).

| Environment | $H{=}16$, $N{=}256$ mean (s) | std (s) | notes |
|---|---|---|---|
| MiniWorld Maze | 0.0463 | 0.0008 | continuous actions |
| Crafter | 0.0493 | 0.0010 | discrete actions |
| DMC-Vision | 0.0478 | 0.0007 | continuous actions |

Table 1: Default planning-call latency. Values are mean $\pm$ std seconds per call.

### F.1 Planning Call Latency

Table 1 reports the mean wall-clock cost of a single imagination-and-score planning call under default settings. Across Maze, Crafter, and DMC-Vision, a planning call is consistently $\approx$50ms.

### F.2 Scaling with Horizon and Candidate Count

Planning cost scales approximately linearly with the rollout horizon $H$: halving $H$ from 16 to 8 roughly halves latency in all environments (e.g., Maze: 0.046s $\rightarrow$ 0.022s; Crafter: 0.049s $\rightarrow$ 0.025s; DMC: 0.048s $\rightarrow$ 0.024s). By contrast, varying the number of candidates $N$ has only a small effect on wall-clock time, because candidates are evaluated in a single batched GPU forward pass. As a concrete illustration, replanning every other step adds $\approx 0.05\text{s} \times 50{,}000 \approx 2{,}500$s ($\sim$42 minutes) per 100k environment steps.

For completeness, Tables 2–4 show the full horizon–choices timing matrices.

### F.3 Meta-policy (PPO) Sequence Processing

The meta-policy is trained using PPO on sequences of length $L$. Table 5 reports the wall-clock cost of processing a batch of sequences for different $L$. At the default $L{=}8$, PPO sequence processing costs $\approx$8–10ms per update, which is small compared to the planning-call cost.

### F.4 Environment Throughput and CPU Bottlenecks (MiniWorld)

We observe that MiniWorld throughput is more CPU-bound than GPU-bound under our setup. On otherwise comparable systems, 350k training steps took 16 hours on an RTX 5090 with an Intel i9-14900K (24 cores, 6.0 GHz max), compared to 24 hours on an RTX 4090 with an i7-13700K (16 cores, 5.4 GHz max), and 36 hours on an RTX 4070 Ti with a Ryzen 9 5900X (12 cores, 4.8 GHz

| H / N | 256 | 128 | 64 | 32 | 16 | 8 | 4 | 2 |
|---|---|---|---|---|---|---|---|---|
| 16 | 0.0463 | 0.0464 | 0.0457 | 0.0445 | 0.0434 | 0.0423 | 0.0419 | 0.0446 |
| 8 | 0.0225 | 0.0230 | 0.0227 | 0.0223 | 0.0216 | 0.0212 | 0.0209 | 0.0212 |
| 4 | 0.0114 | 0.0116 | 0.0115 | 0.0114 | 0.0110 | 0.0106 | 0.0107 | 0.0109 |
| 2 | 0.0058 | 0.0059 | 0.0059 | 0.0058 | 0.0056 | 0.0054 | 0.0054 | 0.0055 |

Table 2: MiniWorld Maze planning-call timings (seconds per call). Means shown; stds are $< 10^{-3}$s except for $N{=}2$ due to measurement noise.

| H / N | 256 | 128 | 64 | 32 | 16 | 8 | 4 | 2 |
|---|---|---|---|---|---|---|---|---|
| 16 | 0.0493 | 0.0495 | 0.0488 | 0.0479 | 0.0462 | 0.0450 | 0.0445 | 0.0471 |
| 8 | 0.0246 | 0.0249 | 0.0244 | 0.0241 | 0.0232 | 0.0228 | 0.0225 | 0.0227 |
| 4 | 0.0123 | 0.0125 | 0.0123 | 0.0122 | 0.0117 | 0.0115 | 0.0114 | 0.0117 |
| 2 | 0.0063 | 0.0064 | 0.0063 | 0.0062 | 0.0060 | 0.0060 | 0.0059 | 0.0059 |

Table 3: Crafter planning-call timings (seconds per call).

max). This suggests that CPU core count and clock speed significantly influence environment-step throughput in these RL environments (we found similar trends with the other two environments).

# G  ABLATION AND SENSITIVITY ANALYSIS

This appendix provides additional ablations and sensitivity sweeps referenced in the main text. We focus on (i) the meta-reward used to train the meta-policy, and (ii) robustness to planner hyperparameters such as the number of candidate rollouts and the commitment (sequence) horizon. Unless otherwise stated, each curve reports the mean with shaded $\pm 95\%$ confidence intervals computed as $2\times$SEM across seeds.

## G.1  META-REWARD (ENTROPY VS. REWARD) ABLATIONS

The main paper reports Crafter meta-reward ablations in Fig. 5a. Here we provide the corresponding ablations on representative DMC-Vision tasks (Fig. 8). We vary the components of $r_{\text{meta}}$ between entropy-only, reward-only, and a 50/50 mixture. As shown in Fig. 8, entropy-only and reward-only yield broadly similar learning trends across tasks, with reward-only often exhibiting slightly higher variance. This is expected given the small number of seeds and the fact that Dreamer's reward predictor can be noisier than the world-model uncertainty signal. The mixed objective is not consistently better and in several cases is slightly worse, suggesting that entropy and reward are only partially aligned in these tasks; mixing them can dilute the planner's anticipatory drive without adding a clearer control signal. Overall, the ablation supports our claim that the method is robust to the entropy reward weighting and does not require careful tuning.

## G.2  SENSITIVITY TO CANDIDATE COUNT

We sweep the number of actor-guided candidates $N$ evaluated per replanning step. Crafter sensitivity to candidate count is shown in the left panel of Fig. 9 (see Fig. 9a), while DMC-Vision sensitivity is shown in Fig. 10. In both regimes, gains persist across a wide span of candidate counts, and the default ($N{=}256$) lies in a stable region. Very small candidate sets (e.g., $N{=}16$) can reduce performance, consistent with the planner having fewer high-return futures to choose among. Because candidates are generated by a greedy actor, trajectories are partially redundant, so moderate oversampling is expected. Importantly, these sweeps align with the timing analysis (Appendix F): increasing $N$ has only a modest impact on wall-clock cost because candidate evaluation is batched on the GPU.

## G.3  SENSITIVITY TO COMMITMENT / META HORIZON

We sweep the commitment horizon (meta PPO sequence length / replanning interval, denoted "seq"). Crafter results are shown in the right panel of Fig. 9 (see Fig. 9b), and DMC-Vision results are shown in Fig. 11. Short-to-moderate commitment typically yields the most reliable gains, matching the main finding that commitment reduces dithering while retaining flexibility to replan. Longer

| $H$ / $N$ | 256 | 128 | 64 | 32 | 16 | 8 | 4 | 2 |
|---|---|---|---|---|---|---|---|---|
| 16 | 0.0478 | 0.0485 | 0.0479 | 0.0467 | 0.0457 | 0.0443 | 0.0461 | 0.0445 |
| 8 | 0.0240 | 0.0243 | 0.0239 | 0.0235 | 0.0228 | 0.0225 | 0.0221 | 0.0224 |
| 4 | 0.0121 | 0.0122 | 0.0121 | 0.0119 | 0.0115 | 0.0114 | 0.0111 | 0.0113 |
| 2 | 0.0062 | 0.0063 | 0.0062 | 0.0061 | 0.0059 | 0.0058 | 0.0057 | 0.0059 |

Table 4: DMC-Vision planning-call timings (seconds per call).

| Environment | $L{=}16$ | $L{=}8$ | $L{=}4$ | $L{=}2$ |
|---|---|---|---|---|
| Maze | 0.0261 | 0.0099 | 0.0092 | 0.0043 |
| Crafter | 0.0234 | 0.0083 | 0.0044 | 0.0041 |
| DMC | 0.0240 | 0.0095 | 0.0048 | 0.0077 |

Table 5: Meta-policy PPO sequence processing time (seconds per update) as a function of sequence length $L$.

commitment can occasionally help (e.g., for smoother dynamics), but is less stable overall. These trends also explain the practical overhead reduction discussed in the timing analysis (Appendix F): longer average commitment amortizes replanning cost over more environment steps.

## H  PLANNER METRICS

This section reports diagnostics that characterise the learned meta-policy and help interpret the practical behaviour of commit-aware replanning. Both plots aggregate quantities per episode and then average across seeds (and environment collections): the solid curve is the mean, the shaded region denotes a single standard deviation, averaged in the same way as the mean but calculated per episode, and the dotted curve shows the average episode-wise maximum, highlighting occasional extreme behaviours, or flexibility. These metrics also connect directly to the compute overhead discussion in Appendix F.

Figure 12 plots the probability that the meta-policy chooses to replan at a given environment step. Across domains, the meta-policy quickly moves away from "always replan" and stabilises at a selective regime: Crafter gradually increases from roughly $0.5$ to around $0.6$–$0.7$ over training, while Maze and DMC stabilise between $\sim 0.6$ and $\sim 0.7$. The dotted average-maximum trace remains close to $1.0$, showing that some episodes briefly approach near-always replanning, but the mean behaviour does not collapse to this regime. Overall, the learned policy maintains the intended commit-aware behaviour while retaining flexibility to replan more aggressively when needed, while proposing different planning regimes for different environments and tasks.

Figure 13 reports the number of environment steps executed before the next replanning event. Commitment lengths remain short on average: Maze and DMC typically commit for about $\sim 2$ steps before reconsidering, while Crafter shows slightly longer commitments early in training ($\sim 2$–$3$ steps) before settling to a similar range. The dotted average-maximum curve, however, remains much higher (around $\sim 8$–$12$ steps depending on domain), indicating that some episodes sustain substantially longer commitments even though typical behaviour favours frequent opportunities to replan.

Appendix F reports an average cost of $\approx 0.05$s per planning call with default settings. Figures 12 and 13 show that replanning occurs on only about half to two-thirds of environment steps, and that each plan is typically executed for multiple steps before replanning. Thus the planning cost is amortised over committed rollouts, reducing the effective overhead per step by roughly a factor of two relative to a worst-case "replan every step" scenario. In expectation this corresponds to an added $\sim 25$–$30$ms per step (about $40$–$50$ minutes per 100k steps), consistent with the wall-clock parity discussion in the timing analysis. Commit-aware replanning therefore provides both behavioural benefits (reduced dithering) and practical throughput gains by avoiding unnecessary imagined rollouts.

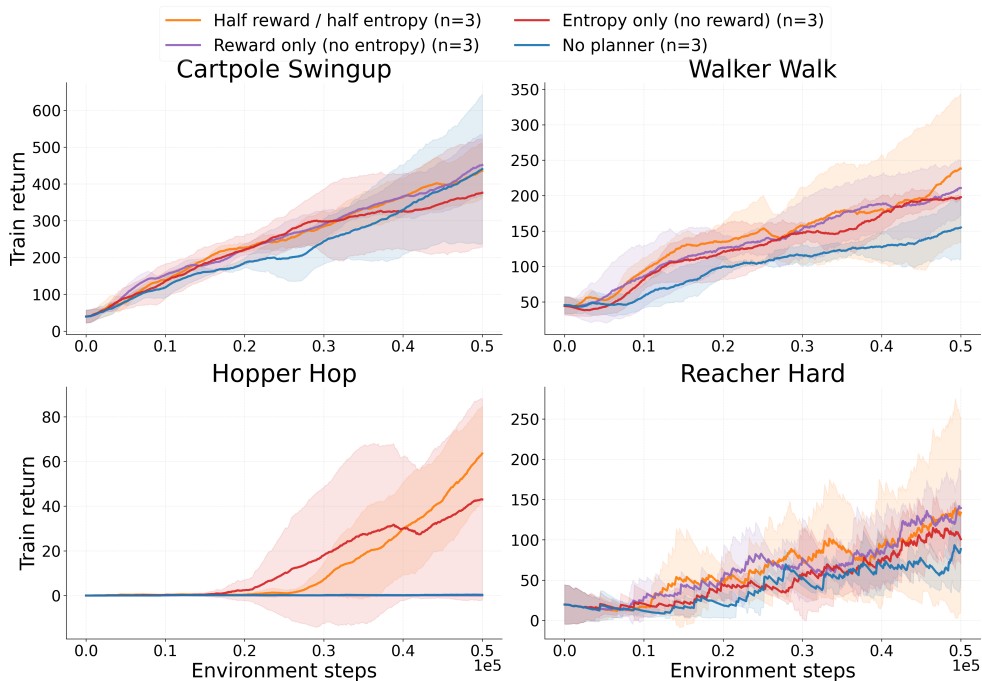

Figure 8: DMC-Vision meta-reward ablation. We vary the meta objective: entropy-only, reward-only, and a 50/50 mixture. Entropy-only and reward-only are comparable (reward-only slightly noisier), while the mixed objective is not consistently better, supporting robustness to $r_{\mathrm{meta}}$ weighting.

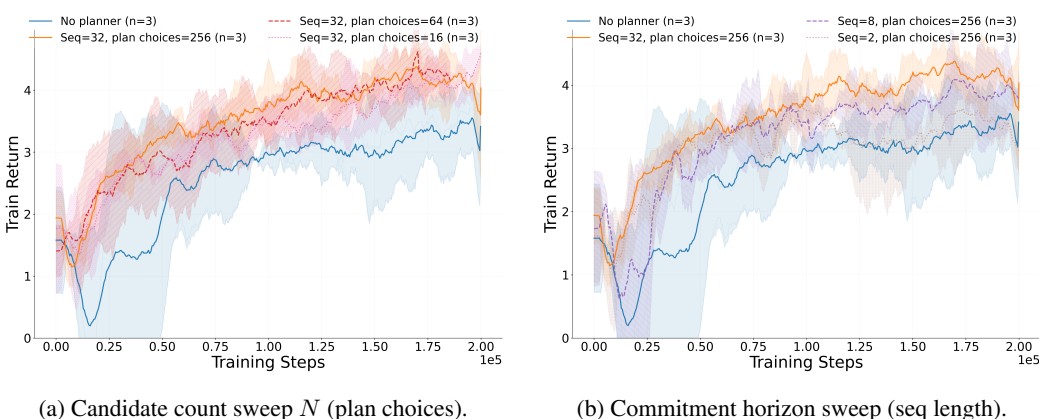

(a) Candidate count sweep $N$ (plan choices).

(b) Commitment horizon sweep (seq length).

Figure 9: Crafter sensitivity sweeps. Left: varying the number of imagined candidates $N$ per replanning step. Right: varying the commitment / replanning horizon (seq). Curves show mean return with shaded $\pm 95\%$ confidence intervals computed as $2 \times$SEM across 5 seeds.

## I   CRAFTER ACHIEVEMENT BREAKDOWN

To better understand where the return gains in Sec. 5.3 come from, we report per-achievement learning curves over the 300k-step budget. Gains are concentrated in routine-forming achievements such as collecting wood, placing tables, and defeating zombies, where short, commit-aware exploratory rollouts appear to reduce dithering and stabilize representation learning (Figs. 14a, 14b, 14c). In contrast, deeper crafting branches (make wooden sword/pickaxe) remain low within 300k steps, especially for the planning variant (Figs. 14g, 14h). We hypothesise this is because the agent attempts these actions early game and finds they do nothing consistently (a nearby table and correct materials

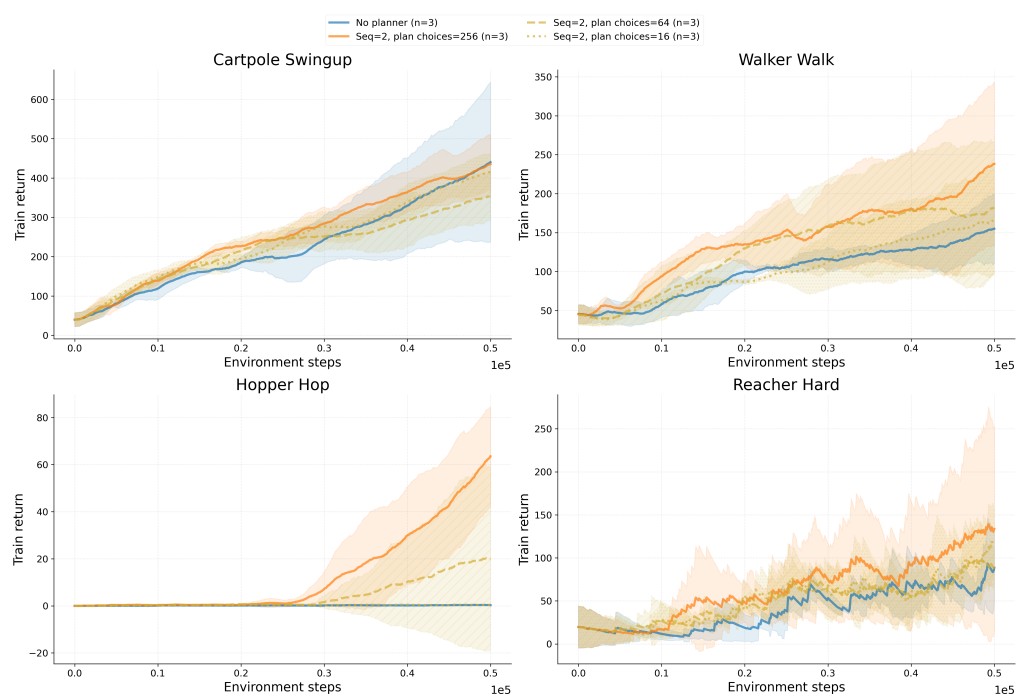

Figure 10: DMC-Vision sensitivity to candidate count $N$ (plan choices) across four tasks (`cartpole_swingup`, `walker_walk`, `hopper_hop`, `reacher_hard`). Each panel reports mean return with shaded $\pm 95\%$ confidence intervals ($2\times$SEM).

in the inventory are prerequisites before the "make pickaxe" or "make axe" button make the respective tools), so the corresponding states become high-confidence low-reward states and are not attempted again, as detailed in failure mode #2. Even so, the planning variant remains better than or competitive with the baseline across the panel (Fig. 14).

Tasks like collect wood and place table are repeatable and reward-dense; plan commitment converts them into habits, yielding steady slopes and higher returns (Figs. 14a, 14b). Combat against zombies sits between routine and opportunistic: once wood/table routines are established, the planner's broader coverage increases encounter rate, so the zombie curve rises earlier and higher than no-plan but still exhibits spikes (Fig. 14c). Making tools remains low for both agents; the planning variant is especially conservative (Figs. 14g, 14h). It is interesting that even though we do not explicitly optimise for reward in the planner, the inherent bias toward rewarding rollouts results in what is effectively zombie and tree farming behaviour (Figs. 14a, 14c). Collect drink, collect sapling and eat cow (Figs. 14d, 14e, 14f) are all roughly matched between the no plan variant and the planning variant, indicating these routine-light behaviours benefit less from commitment.

## J DEFAULT CONFIGURATION AND CODE BASE

### J.1 DEFAULT CONFIGURATION

The following listing provides the default hyperparameters and settings used in our experiments.

```
use_plan: True

logdir: null
traindir: null
evaldir: null
offline_traindir: ''
offline_evaldir: ''
seed: 0
```

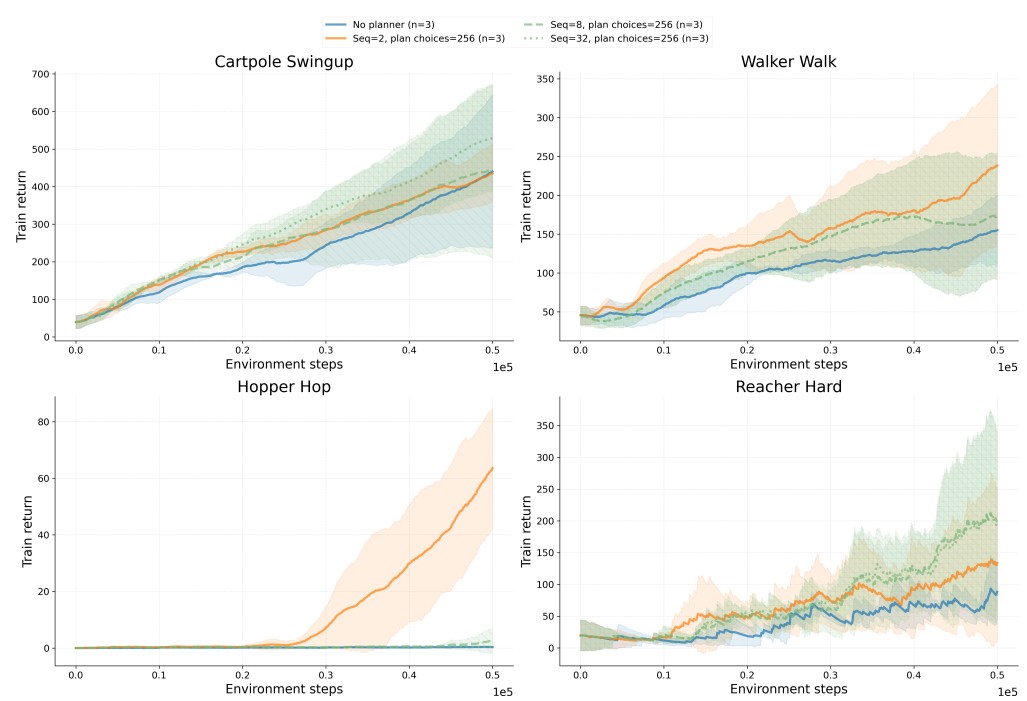

Figure 11: DMC-Vision sensitivity to commitment horizon (seq length) across four tasks (`cartpole_swingup`, `walker_walk`, `hopper_hop`, `reacher_hard`). Each panel reports mean return with shaded ±95% confidence intervals (2×SEM).

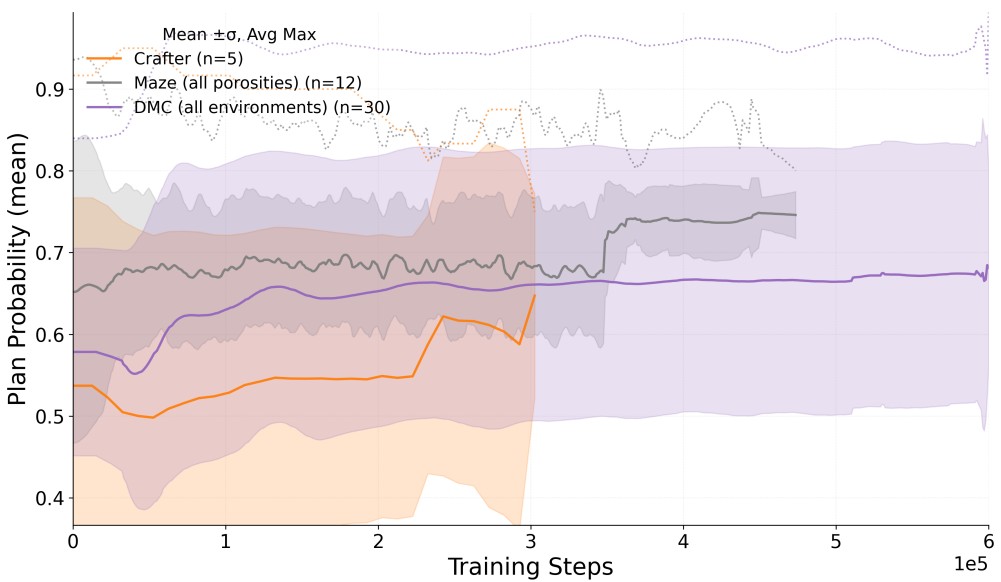

Figure 12: Planning probability over training. Solid: mean across seeds; shaded band: 1 standard deviation averaged; dotted: average episode-wise maximum. The meta-policy settles into a selective replanning regime, with replan probabilities typically between $\sim 0.5$ and $\sim 0.7$ rather than replan-every-step behaviour.

```
deterministic_run: False
steps: 1e6
parallel: False
```

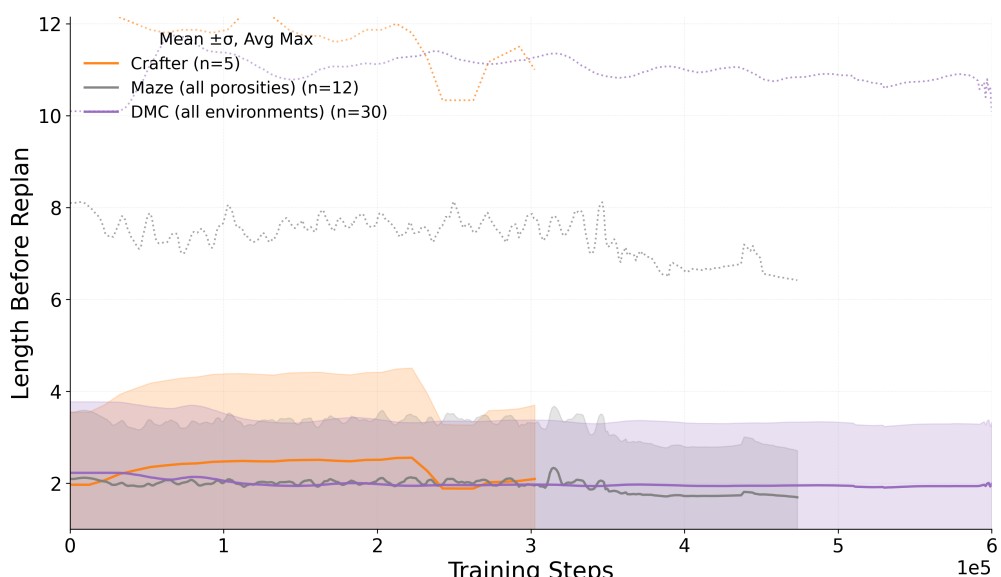

Figure 13: Length before replanning. Solid: mean across seeds; shaded band: 1 standard deviation averaged; dotted: average episode-wise maximum. The planner usually commits for $\sim$ 2–3 steps, with occasional long commitments.

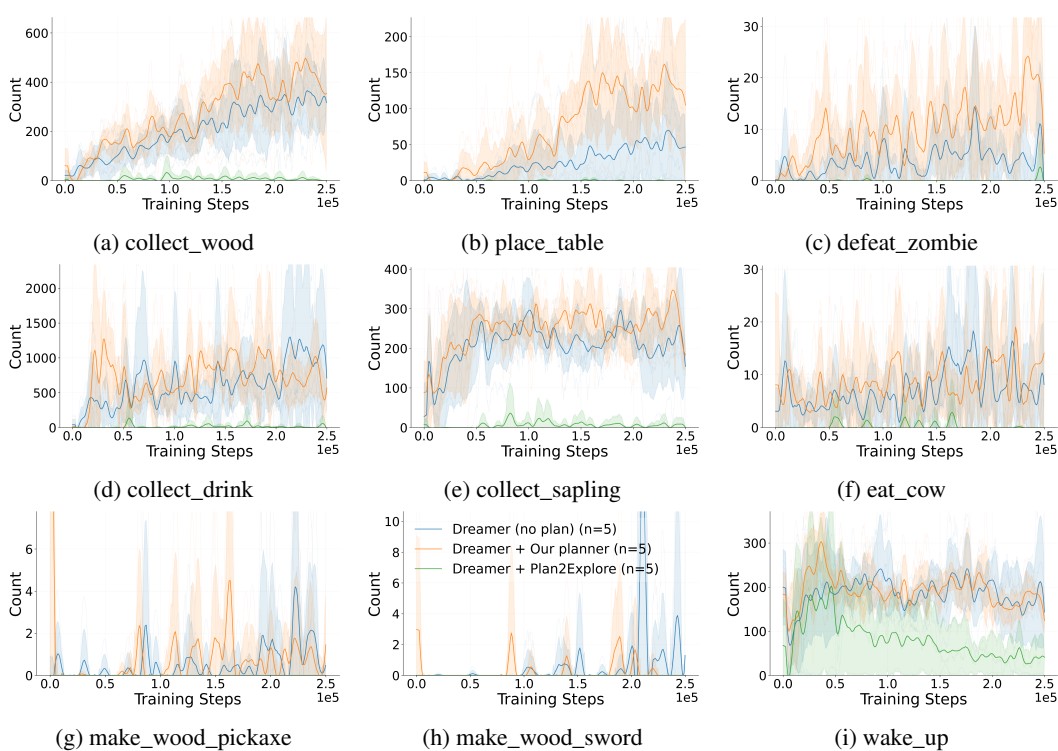

(a) collect_wood

(b) place_table

(c) defeat_zombie

(d) collect_drink

(e) collect_sapling

(f) eat_cow

(g) make_wood_pickaxe

(h) make_wood_sword

(i) wake_up

Figure 14: Crafter achievement counts over 300k steps (3 seeds) for no plan vs the planning variant vs plan2explore. The planning agent forms reliable routines (wood, table, zombie) and improves sample efficiency, while deeper crafting remains conservative within this budget.

```
eval_every: 1e4
eval_episode_num: 10
log_every: 1e4
```

```
reset_every: 0
device: 'cuda:0'
compile: True
precision: 16
debug: False

# Environment
task: 'dmc_walker_walk'
size: [64, 64]
# envs: 1
# action_repeat: 1
time_limit: 1000
grayscale: False
prefill: 2500
reward_EMA: True

# Model
dyn_hidden: 512
dyn_deter: 512
dyn_stoch: 32
dyn_discrete: 32
dyn_rec_depth: 2
dyn_mean_act: 'none'
dyn_std_act: 'sigmoid2'
dyn_min_std: 0.1
grad_heads: ['decoder', 'reward', 'cont', 'entropy']
units: 512
act: 'SiLU'
norm: True
encoder:
  {mlp_keys: '$^', cnn_keys: 'image', act: 'SiLU', norm: True, cnn_depth: 64, kernel_size: 4,
decoder:
  {mlp_keys: '$^', cnn_keys: 'image', act: 'SiLU', norm: True, cnn_depth: 32, kernel_size: 4,
actor:
  {layers: 2, dist: 'normal', entropy: 3e-4, unimix_ratio: 0.01, std: 'learned', min_std: 0.1,
Q:
  {layers: 2, dist: 'symlog_disc', slow_target: True, slow_target_update: 1, slow_target_fract
critic:
  {layers: 2, dist: 'symlog_disc', slow_target: True, slow_target_update: 1, slow_target_fract
reward_head:
  {layers: 2, dist: 'symlog_disc', loss_scale: 1.0, outscale: 1.0}
entropy_head:
  {layers: 2, dist: 'symlog_disc', loss_scale: 1.0, outscale: 1.0}
cont_head:
  {layers: 2, loss_scale: 1.0, outscale: 1.0}
dyn_scale: 0.5
rep_scale: 0.1
kl_free: 1.0
weight_decay: 0.0
unimix_ratio: 0.01
initial: 'learned'

# Training
batch_size: 16
batch_length: 64
train_ratio: 512
pretrain: 100
model_lr: 1e-4
opt_eps: 1e-8
grad_clip: 1000
dataset_size: 1000000
opt: 'adam'

# Behavior.
discount: 0.997
```

```
1296    discount_lambda: 0.95
1297    imag_horizon: 15
1298    imag_gradient: 'dynamics'
1299    imag_gradient_mix: 0.0
1300    eval_state_mean: False
1301
1302    # Exploration
        expl_behavior: 'greedy'
1303    expl_until: 0
1304    expl_extr_scale: 0.0
1305    expl_intr_scale: 1.0
        disag_target: 'stoch'
1306    disag_log: True
1307    disag_models: 10
1308    disag_offset: 1
1309    disag_layers: 4
        disag_units: 400
1310    disag_action_cond: False
1311
1312    # plan_behavior:
1313    plan_max_horizon: 16
1314    plan_choices: 256
1315    plan_train_every: 32
        sub_batch_size: 64
1316    num_epochs: 30
1317    buffer_size: 32768
1318    clip_epsilon: 0.2
1319    gamma: 0.99
        lmbda: 0.95
1320    entropy_eps: 0.1
1321    num_cells: 256
1322    lr: 0.003
1323    seq_length: 8
1324    buffer_minimum: 512
        meta_action_quant: 5                    # used in CategoricalSpec
1325    num_meta_action_lwr: 2                     # used in CategoricalSpec
1326    ent_multiplier: 1.0                      # multiplier for entropy in _flow method
1327    rew_multiplier: 1.0                      # multiplier for reward in _flow method
1328
1329    dmc_vision:
1330      steps: 1e6
          action_repeat: 2
1331      envs: 1
1332      train_ratio: 512
1333      video_pred_log: false
1334      encoder: {mlp_keys: '$^', cnn_keys: 'image'}
          decoder: {mlp_keys: '$^', cnn_keys: 'image'}
1335
1336    crafter:
1337      task: crafter_reward
1338      step: 1e6
1339      action_repeat: 1
          envs: 1
1340      train_ratio: 512
1341      video_pred_log: false
1342      dyn_hidden: 1024
1343      dyn_deter: 4096
1344      units: 1024
          encoder: {mlp_keys: '$^', cnn_keys: 'image', cnn_depth: 96, mlp_layers: 5, mlp_units: 1024}
1345      decoder: {mlp_keys: '$^', cnn_keys: 'image', cnn_depth: 96, mlp_layers: 5, mlp_units: 1024}
1346      actor: {layers: 5, dist: 'onehot', std: 'none'}
1347      value: {layers: 5}
1348      reward_head: {layers: 5}
1349      cont_head: {layers: 5}
          imag_gradient: 'reinforce'
```

### J.2 CODE BASE

Our implementation is forked from `https://github.com/NM512/dreamerv3-torch/blob/main/dreamer.py`. We adapt this code to our setting while retaining the default configuration listed above.

