# OpenReview forum: "ENTER THE VOID: EXPLORING WITH HIGH ENTROPY PLANS"
_ICLR.cc/2026/Conference — Submitted to ICLR 2026_

### Official Review · Reviewer_WShA · 2025-10-20

**Soundness:** 1
**Presentation:** 3
**Contribution:** 1
**Rating:** 2
**Confidence:** 3

**Summary:**

The authors propose to train Dreamer(/like) world-model-based-agents by acting with an exploration policy that is inferred in real time by planning with the world model.
The main contribution is the usage of latent-transition-model entropy to direct the policy towards novel states.

**Strengths:**

The methods seems rather simple.

Presentation is rather clear.

I like the direction that makes the planner adaptive, deciding when to spend compute for replanning, and when to simply follow the existing plan.

**Weaknesses:**

Soundness:
1. It is not clear to me that the method is sound. In stochastic environments, maximizing the sum of individual transition entropies amounts to searching for the most noisy trajectories, which is opposed to exploration, which should look for novel transitions. The authors acknowledge this in lines 176-180.
2. In det. environments, the entropy of the learned transition model can be used as a pseudo-metric for the novelty of the state. If we limit this method to det. environments, it's not clear to me why we should take the entropy instead of the variance, where the variance directly produces theoretically motivated and well investigated UCB exploration (see [1]).
2. Even in det. envs., It's not clear to me why the entropy of the learned transition model should be a better metric for novelty than the disagreement of the model ensemble (i.e. plan2explore). This seems to me to suggest problems in the evaluation setup. Can the authors explain why plan2explore can be expected to fail in maze? Isnt it essentially the same method, only operating based on a theoretically sound metric for novelty (ie a metric that can seperate aleatoric from epistemic uncertainty)?
3. The method seems unnecessarily myopic. Why not learn the intrinsic-reward (the entropy in this case) in a value function, train a policy with respect to this value function, and produce an agent that is able to explore in the direction of novel states, even if these states are not reached in the rollout-length? This is a rather standard method, which again amounts simply to UCB-exploration (see [1] for more details).

Empirical evaluation:
1. A very small number of seeds is used in evaluation, resulting in no statistical significance of the results in most enivornments.
2. Stat. signif. is measured as STD. I don't think STD is the right metric, as it relates to the deviation between the seeds, not the deviation of the mean (ie stat. signif. of the performance gain). I would suggest 2 SEM (i.e. ~95% Gaussian CI).
3. Missing baselines: [1] provides a structured and better theoretically motivated method for planning-for-exploration in MBRL, which is applicable to Dreamers. With the same entropy based estimates for epistemic uncertainty, is there any reason to use the planner proposed in this paper over [1]?
4. Presentation: I would suggest changing the legends to the following: *PPO*, *Dreamer* (previously "no-plan"?), *Dreamer + Our planner* (previously "using plan"), *Dreamer + Plan2Explore* (or P2E for short) for clarity, to make it easier to follow from a glance what is actually compared.

[1] Oren, Yaniv, et al. "Epistemic Monte Carlo Tree Search." ICLR 2025.

**Questions:**

Comments:
1. I dont understand (/agree with) the seperation of intrinsic reward into "retrospective and anticipatory" methods. If we have access to the model, the one "correct" (or "complete) way to take intrinsic-reward into account is by learning the intrinsic reward inside a value function (retrospective?) and using planning / search to combine the value predictions and the intrinsic reward predictions (anticipatory?) to find an even-better-for-exploration action. If we don't learn the reward in a value function, it seems to me we are myopic - ie suboptimal - for no reason. If we don't have access to the model, we can only use the value function that learned the intrinsic reward. So isn't the better seperation between model-based and model-free methods (i.e. planning vs. no-planning?).
2. line 088: MCTS has been extended to stochastic and cont. envs. (sampled MZ and stochastic MZ [1,2]).
3. Missing reference for PPO (line 049).
4. Line 096 towards -> toward.

[1] Hubert, Thomas, et al. "Learning and planning in complex action spaces." International Conference on Machine Learning. PMLR, 2021.

[2] Antonoglou, Ioannis, et al. "Planning in stochastic environments with a learned model." International Conference on Learning Representations. 2021.

---

> ### Author Response · Authors · 2025-11-27
>
> W1: We appreciate the reviewer’s concerns and have revised Section 4.1 to make our assumptions explicit. We do not claim prior entropy is a perfect epistemic measure, we instead treat it as a proxy for anticipated posterior correction induced by the KL term in Dreamer’s RSSM. While naively optimising for just entropy of state distribution would amount to searching for noise, we instead pre select from actor-guided rollouts (thereby all candidates are distinctly not noise) and we then use a shaped reward that combines environment return and latent prior entropy over a horizon L(Eq. (6)) to select the most informative, task-relevant future.
>
> W2: We agree that variance-based UCB and ensemble disagreement are principled epistemic proxies. Our main reason to use entropy is compatibility with both discrete and continuous RSSMs: entropy applies uniformly to Gaussian and factorised categorical latents, whereas variance is tied to the Gaussian case. Section 4.1 now spells out this design choice.
>
> W3: We also include Plan2Explore as an ensemble-disagreement baseline in all domains (Section 5). In our mazes and most DMC-Vision tasks, Plan2Explore underperforms; Section 5.1 and Appendix H hypothesise that this stems from frequent replanning without commitment, whereas our meta-policy explicitly learns when to commit vs. replan. It is interesting to note that Plan2Explore has slower convergence than base dreamer as shown by their own paper - it could be possible that plan2explore simply hasn’t started being successful in our given step budget. Furthermore, ensemble disagreement is intuitively poorly suited to a maze – ensembles will frequently disagree about which direction to take at ‘crossroad states’ and mazes are full of these. Ensemble disagreement is also computationally expensive, plan2explore mitigates this by using disagreement of shallow proxy networks, which is why it will not have the same benefits as using a committee of full dreamer world models.
>
> W4: Our planner is not purely one-step: it maximises a multi-step sum of latent entropy along each candidate rollout of length H=16 and trains the meta-policy on sequences of length L=32(Section 4.2), so the signal is explicitly non-myopic. Section 2.1 has been rewritten to clarify our taxonomy: we distinguish retrospective intrinsic rewards (value heads trained from replay) from anticipatory methods that score imagined futures; this axis is indeed orthogonal to model-based vs. model-free. We now state that learning an additional intrinsic-value head on top of our planner is a promising direction for future work, but comes with additional parameters and lag before the signal becomes reliable.
>
> W5, W6, W8: We now run 5 seeds for MiniWorld and DMC and 5 seeds for Crafter, and we adopt the reviewer’s suggestion of reporting 95% confidence intervals (2×SEM) throughout all graphs presented in the paper. Our results now look better too, with some separation between confidence intervals. We have also implemented your recommendations regarding the legends. They are now clearer and easier to follow!
>
> W7: We agree that Epistemic MCTS is an interesting complementary approach. However, the current EMCTS formulation targets low-dimensional, discrete-state domains with value-uncertainty estimates, whereas our setting uses Dreamer’s high-dimensional latent RSSM with continuous actions and pixel observations. Adapting EMCTS would require non-trivial architectural changes (tree search in latent space, value-uncertainty heads, continuous-action branching) beyond the scope of this work.
>
> Q1: We agree that model-based vs. model-free (planning vs. no-planning) is an important axis, and we now explicitly state that our “retrospective vs. anticipatory” distinction is orthogonal to this. Our goal with this terminology is to separate methods that (i) derive novelty signals from past data via an intrinsic-value head trained on replay (“retrospective”), from those that (ii) compute novelty directly on imagined futures using the current world model (“anticipatory”). Both can coexist in a model-based agent.
>
> In our setting, we already learn the external reward value as in Dreamer, but we deliberately avoid adding a second intrinsic-value head that must be trained from scratch to predict model error or ensemble disagreement. Instead, we exploit the fact that the world model can provide short-horizon uncertainty signals “for free” via its latent dynamics, and we combine these with reward in our planner objective. This avoids the additional parameters and lag before an intrinsic value network becomes reliable, while still yielding non-myopic exploration through multi-step imagined rollouts and plan commitment. Curiosity value-heads for UCB-style exploration are therefore complementary to our anticipatory planner, not mutually exclusive; we view integrating both as an interesting avenue for future work.
>
> Q2, Q3, Q4: We have made all recommended corrections.

---

> ### Comment · Reviewer_WShA · 2025-11-27
>
> I thank the authors for the detailed rebuttal.
>
> 1. **Changes:** please mark changes from previous drafts clearly in the document (perhaps in a different color), so that the reviewers do not have to-reread unnecessary sections, or wonder what changed.
>
> 2. **W1:** It is not clear to me what "pre select from actor-guided rollouts (thereby all candidates are distinctly not noise)" means exactly. In addition, isn't incorporating the reward and the entropy the same as doing intrinsic reward, with a not-necessarily-correct measure of epistemic uncertainty (as the authors mention, this entropy can simply be a direct result of noise)?
>
> 3. **W2:** Why would variance be tied to the Gaussian case?
>
> 4. **W3:** By going with the ensemble disagreement as a metric for epistemic uncertainty, it seems to me that plantoexplore should correctly identify novel states and direct itself towards them (states with high disagreement - novel states). It remains unclear to me why it would fail in these case, especially compared to the baseline dreamer.
>
> 5. **W4:** It is then myopic in length L=32, correct? Unlike a value-uncertainty-head, which is truly not-myopic?
>
> 6. **W5-6-8:** I thank the authors for the changes. In my experience with DM control, at least 20 seeds are necessary for reliable results, as the learning behavior of the agents in these environments can be very unstable. 5 seeds are not a sufficient number of seeds, in my opinion. I note that in Figure 2, all confidence intervals between Dreamer and Dreamer + Contributions overlap.
>
> 7. **W7:** Tree search in latent space is by now an old and extremely successful approach, see MuZero [1]. Value uncertainty heads should, in my opinion, be incorporated into the authors' method to start with: otherwise, it seems to me it will remain unnecessarily myopic to length L, the planning horizon (W4). Continuous action branching in (/latent space) tree search has been addressed by SampledMuZero [2]. Since all of those are by now old baselines (2020-2021), it does not seem to me to go beyond the scope of this work.
>
> 8. **Q1**: I think that both *should* coexist in model based methods, otherwise the method is unnecessarily myopic in a truncated horizon. Wouldn't you agree?
>
> Finally, as an alternative structured way to incorporate epistemic uncertainty into planning, I think Epistemic MCTS should be mentioned in the related work.
>
> [1] Schrittwieser et al. "Mastering atari, go, chess and shogi by planning with a learned model." Nature 2020.
>
> [2] Hubert, Thomas, et al. "Learning and planning in complex action spaces." ICML, 2021.

---

> ### Author Response · Authors · 2025-11-27
>
> Changes marking
> - We will upload a revised manuscript where all modified passages are clearly marked.
>
> W1
> - At each environment step we first sample candidate trajectories using Dreamer's world model under it's greedy
> actor, and then score those trajectories by a mixture of predicted reward and latent prior entropy. This means we are not sampling arbitrary high-entropy transitions: all candidates already correspond to trajectories that the actor
> expects to be at least reasonably reward-seeking, and entropy is used to break
> ties among task-relevant trajectories rather than to chase pure noise.
>
> W2
> - We have updated our original phrasing. In the categorical case used by DreamerV3, there is no intrinsic, model-provided scalar variance per latent dimension: any “variance-like’’ quantity would require introducing an extra head or choosing an arbitrary numeric embedding of the categories. Entropy, by contrast, is defined directly from the distribution itself and can be computed from both Gaussian parameters and categorical logits. This makes entropy a natural, head-free uncertainty proxy across both Gaussian and factorised categorical RSSMs. We have updated Section 4.1 accordingly to remove the imprecise wording.
>
> W3
> - We agree that, in principle, Plan2Explore should be able to seek entropy. However, it doesn’t have good sample efficiency and this does not surprise us, as their original paper shows that they converge slower than base dreamer.
>
> This is because Plan2Explore uses disagreement to steer towards novel states looking ahead just one step, myopically. In our mazes, we theorise that this leads to frequent changes of direction at junctions because these are exactly where the ensemble disagrees most. Our meta-policy, by contrast, learns when to commit to a chosen rollout, reducing dithering.
>
> Furthermore, Plan2Explore uses a small ensemble of shallow proxy networks (instead of training and maintaining a committee of full dreamer models) and measures their disagreement. While much cheaper computationally it also means that the epistemic signal is only loosely coupled to the main RSSM that trains the actor. In our experiments, this signal appears sufficient for some underactuated control tasks (e.g., acrobot_swingup, where Plan2Explore is indeed strong; Fig. 2e) but less effective in high-dimensional navigation with partial observability.
>
> W4, Q1, W7
> - Yes, our meta-planner is finite-horizon - it aggregates reward and entropy over a sequence of length L. This is exactly analogous to other predictive control methods that also optimise over a truncated horizon. We agree that an intrinsic value head can propagate uncertainty beyond this horizon and be less myopic. However, we explicitly set out to avoid learning an additional intrinsic value network with its own optimisation dynamics, uncertainty, and prediction noise. Instead, we use a quantity that the world model already maintains (the prior’s predictive entropy) as an inference-time signal.
> We also agree with the reviewer that ideally epistemic signals from value-uncertainty heads and short-horizon planning should coexist rather than compete, we now say this explicitly in Section 2.1 and Section 4.2. We see our work as a first step that isolates the benefit of commit-aware, entropy-guided planning on top of an existing world model; integrating a full value-uncertainty head is a substantial extension that deserves its own careful study (architectural choices, targets, regularisation, and compute budget).
>
> W5, W6, W8
> - The original Dreamer paper only uses five seeds in the DMC environments and we felt that the cost of any more seeds to be expensive; the cost of 15 more seeds to be prohibitively so. Our current setup uses 5 seeds for DMC-Vision and MiniWorld and 5 for Crafter, which is a compromise between robustness, breadth of environments, and available compute.
> Our DMC-Vision testbench is 6 tasks × (3 methods + 6 ablations) x 5 seeds. It takes 8h wall-clock per trace per seed on a high-end GPU (Sec. 5, Appendix F). Scaling this to 20 seeds per task would require on the order of 10^3GPU-hours.
> Figure 2 shows significant separation of confidence intervals on hopper hop and walker walk. DMC is inherently a control task where long term planning and commitment only increase the sample efficiency for some of the more complex tasks.
>
> W7
> - We agree that MuZero-style tree search and its extensions are powerful and relevant. However, integrating such tree search into our method would require non-trivial architectural choices (an explicit search policy and continuous-action branching) and would greatly increase per-step planning cost beyond our current batched rollouts (Appendix F).

---

### Official Review · Reviewer_KowQ · 2025-10-31

**Soundness:** 2
**Presentation:** 1
**Contribution:** 3
**Rating:** 2
**Confidence:** 3

**Summary:**

The paper proposes an extension of the Dreamer model-based RL method to improve exploration of rarely visited states. The authors propose to generate $C$ standard (i.e. greedy) trajectories and then use the ones with the highest entropy of the transition predictor (prior) for training the policy. They also propose another meta-controller, trained to maximize both reward and entropy, that decides randomly whether this process shall be used or not. Experiments on DMC show in 2 out of 6 environment a significant advantage, but on 9 Crafter environments there is no significant improvement observable.

**Strengths:**

The topic is very interesting. Last ICLR another paper [1] showed impressive improvements in exploration when AlphaZero plans optimistically w.r.t. epistemic uncertainty of the involved neural networks, but only showed minor to no improvement for learned models like MuZero (in the appendix). Better exploration with a Dreamer algorithm (which learns the model) would be of great interest to the community, and the presented idea of using internal prior entropy as a stand-in of epistemic uncertainty is (to the best of my knowledge) novel and interesting.

**References**

[1] Oren et al. (2025). "Epistemic Monte Carlo Tree Search". Proceedings of the International Conference on Learning Representations. URL https://arxiv.org/abs/2210.13455

**Weaknesses:**

I recommend to reject the paper in its current form, because (i) incomplete or missing formalism, description and intuition, (ii) insufficient distinction between aleatoric and epistemic uncertainty, (iii) insufficient reasoning why the prior neural network should produce reliable entropy estimates for unseen states, and (iv) results that are inconclusive. In detail:

1. I have read section 3 and 4 multiple times, and I am still not sure what the authors exactly propose to do. This is partly because the formalism is incomplete. Section 3 needs a clear introduction of MDP and POMDP. Much of the math in Section 4 is incomprehensible (at least to me). For example: I assume the idea is that IG in Equation 3 is minimized by RSSM training (as the "KL divergence loss can also be interpreted as the model's information gain"). Large prior entropy means large prior standard deviations $\sigma_p$ (for Gaussians), leading to *smaller* IG. But how does choosing trajectories (or more precisely random outcomes in the trajectories) that artificially lower the IG incentivize the agent to explore less visited states in the environment? What is the max operation in Equation 5 over? To "choose the [trajectory] with the highest entropy"? This biases the imagined environment, but how does it affect the choice of actions in the environment towards epistemic uncertainty?
Another example of unclear formalism is the following discussion on $\hat p$, which conditions on different terms every time it is mentioned. What is this distribution? A thought experiment or something that is learned somewhere? What are the states $s_t$ that is reasoned over, the true environment states? But those do not depend on $h_{0:t}$. Finally, I did not understand why a meta-planer is even necessary. Assuming that the authors reasoning on lowering the IG in correct, why stochastically do it only every now and then? Why not do it all the time (called MPC style later)?
2. In the second half of Section 4.1 the authors seem to develop an argument  when the prior entropy is a good stand-in for epistemic uncertainty (I believe) and when not. However, I find it hard to understand why this should be the case in the first place. As the authors note, high prior entropy will *after sufficient training* on a data set correspond to high aleatoric uncertainty. But why should it be related to epistemic uncertainty about the agent's knowledge of the environment? This is what exploration (in the environment) should be based on, as only epistemic uncertainty is reducible with more samples. Or did I misunderstand what the authors call "exploration"?
3. All presented arguments only hold if the neural network produces the "correct" prior entropy. This will only be the case in or near the training set. However, exploration is about finding states that have *not* been explored, that is, which are not in (or even near) the training set. So why do the authors expect the prior entropy to be a good exploration signal? The paper misses a discussion on this.
4. While there are a lot of experiments, and the discussion of the results often contextualizes them correctly ("gains are modest but consistent"), it must be observed that only in ```walker_walk``` and ```hopper_hop``` the proposed methods shows a significant advantage over the "no plan" baseline.  In particular the ```Crafter``` experiments do not show **any significant** improvement. Given that the paper's motivation is somewhat hard to follow (at least for me), this is not a sufficient level of evidence to claim that the proposed method has any systematic effect whatsoever. Without clearly separated standard deviations (or a more rigorous measure of statistical significance), none of the paper's claims can be relied on.

**Questions:**

- See the questions above.
- What does the "state" in Equations 4 and 5 refer to? The input of the entropy $h_t$?

---

> ### Author Response · Authors · 2025-11-27
>
> We thank the reviewer for the detailed comments. We have substantially revised the paper to address (i) missing formalism, (ii) the entropy/epistemic discussion, and (iii) the experimental presentation.
>
> W1: We now introduce the MDP/POMDP formalism explicitly in Appendix B and reference it in Section 3 to make the assumptions behind Dreamer and our planner precise. Sec. 4.1 has been rewritten to (a) present the KL/IG expression (Eq. 3), (b) explain that we use IG purely as an interpretive view of the KL term rather than an additional loss, and (c) motivate prior entropy as a scalar statistic of the predictive prior that increases when the model expects large posterior corrections.
>
> We clarify that the “state” in Eqs. (4–5) is the latent prior $z ̂_t∼p_ϕ (z_t∣h_t)$, not the true environment state, and that the maximisation in Eq. (5) is over a finite set of actor-guided imagined rollouts: at each replanning step we select the candidate trajectory whose cumulative prior entropy is largest and execute its prefix in the real environment. This is now spelled out in Sec. 4.2 and in Algorithm 1 in Appendix C.
>
> The role of the meta-planner is also clarified: Sec. 4.2 now explains that it learns when to replan vs. commit so as to avoid myopic “replan every step” behaviour and amortise planning cost. Sec. 5.4 and Fig. 4 compare this to an MPC-style variant that replans at every step, showing that learned commitment yields both better performance and better efficiency than MPC-style replanning.
> W2, W3: Sec. 4.1 now explicitly treats prior entropy as a proxy for epistemic uncertainty, not a perfect decomposition. We discuss two concrete failure modes: (i) multi-modal aleatoric transitions, where a unimodal RSSM prior must “cover” several outcomes and thus yields inflated entropy, and (ii) rare latent branches whose uncertainty can remain underestimated. We do not claim to fully separate epistemic and aleatoric uncertainty (which is known to be hard); instead, we mitigate aleatoric inflation by only scoring actor-guided, reward-seeking candidate rollouts. Under an actor trained to maximize the return, repeatedly visited noisy states tend not to dominate the candidate set; poorly modelled or under-explored regions (subsequently with high entropy) that have potential for high rewards do.
>
> Because the prior is a deterministic function of the belief state learned by the RSSM, its entropy is highest precisely where the model expects strong posterior corrections. We therefore use prior entropy as a model-grounded, reward-free heuristic for anticipated model error, not as a learned intrinsic reward head with its own optimisation dynamics. We have clarified this perspective in the text.
>
> W4: All learning curves now report means with 95% confidence intervals (2×SEM) across seeds, and we have moved the per-achievement Crafter breakdown to Appendix I as it was causing confusion. The revised plots show that our planner consistently improves sample efficiency: on MiniWorld it reduces episode length across porosities; on DMC it yields modest but reliable gains on 4/6 tasks; and on Crafter it achieves ≈20% higher average return and reaches comparable thresholds in roughly half the environment steps of base Dreamer within the fixed 300k-step budget. We do not claim to change the world model; we simply frame the method as a robust, inference-time exploration layer that improves data efficiency when anticipatory entropy-guided planning is beneficial.

---

> > ### Comment · Reviewer_KowQ · 2025-11-27
> > **I still do not see enough evidence**
> >
> > I thank the authors for their rebuttal and attempts to address my concerns. However:
> > - It is not sufficient to introduce the notation used in a paper in an appendix, as papers need to understandable on their own.
> > - Could you please mark the changes in the paper with a different color? I currently cannot see some of the claimed changes (like that the maximum in eq.5 is over imagined rollouts--the word "rollout" does not appear in Section 4.1).
> > - Why would you expect that "earned commitment yields both better performance and better efficiency"? Is there a cost (in performance, not just in time) to making rollouts at every time step?
> > - Looking at Figure 2, in none of the improvements have separated 95% confidence intervals. The claimed "consistently improves sample efficiency" are not a statistical test. One could accept them as an indication that your method might improve performance, but not as evidence that it does.
> >
> > In particular the unclear evidence does not allow me to increase my rating.

---

> > > ### Author Response · Authors · 2025-11-27
> > >
> > > We thank the reviewer for their prompt reply.
> > >
> > > 1) - Noted, we have added the requested formalism to the beginning of the preliminaries section.
> > >
> > > 2) - We will shortly provide a highlighted copy, and you can find all rollout related content in section 4.2.
> > >
> > > 3) - The meta-planner was introduced to avoid myopic “replan at every step” behaviour, which we found empirically to cause dithering: actions fluctuate in response to short-horizon predictions and sensor noise, instead of following short but coherent multi-step behaviours. In Section 5.4 we compare our commit-aware planner to an MPC-style variant that always replans at every step. Under matched environment-step budgets, the MPC-style variant converges to worse task performance than our learned-commitment planner in both Maze (longer episodes) and Crafter (lower returns), despite having more frequent replanning. We will make this performance gap explicit in the text (beyond the curves already shown in Fig. 4) and clarify that the benefit of commitment is not just in wall-clock efficiency but in avoiding behavioural dithering that degrades task return.
> > >
> > > 4) - We understand the reviewer’s concern about overlapping confidence intervals. Pixel-based Dreamer-style agents are known to exhibit substantial run-to-run variability, which leads to relatively wide confidence bands even for strong baselines. In our experiments, we nevertheless observe that the planning variant matches or exceeds base Dreamer on most tasks, while Plan2Explore sometimes underperforms and can reduce performance relative to the baseline. Across the full suite of experiments we ran, we did not observe systematic degradations from adding the planner, and the planner achieves higher sample efficiency in well over half of the individual task settings. Although these improvements are not always statistically separated at the 95% level, they provide consistent empirical evidence that inference-time entropy-guided planning can be beneficial. We have softened the wording in the manuscript accordingly, framing our findings as indicative trends rather than definitive statistical guarantees.

---

### Official Review · Reviewer_WBUJ · 2025-11-01

**Soundness:** 2
**Presentation:** 3
**Contribution:** 4
**Rating:** 4
**Confidence:** 3

**Summary:**

Model features an exploration method for model-based reinforcement learning from high-entropy exploration. Authors propose to focus on exploring the states with high-entropy stochastic part in learned state representation. To do so, authors use a train-time planing:
- In planing stage, algorithm generates N imaginary trajectories using low-level actor (that is  learned as in vanilla Dreamer V3), chooses one with highest cumulative entropy over learned hidden state representation's stochastic parts.
- Throughout episode, algorithm uses high-level PPO-based agent which decides wether we should change plan at current step or continue execution.

**Strengths:**

- Paper proposes a novel exploration algorithm for model-based reinforcement learning, applying it to Dreamer V3. Proposed algorithm generalizes well as long as learned world model uses RSSM as backbone
- Aside from proposition of exploration target, paper proposes usage of PPO-based high-level trainable controller that dynamically decides when to change current plan during training and replanning based on imaginary trajectories that generated with low-level RL agent. While planning is not novel for MBRL itself, such hybrid approach may have high potential in future research and can be applied to other exploration techniques in model-based setup.

**Weaknesses:**

- Weak experimental base. In past years several model-based exploration techniques were published (i.e. [1](https://proceedings.neurips.cc/paper/2021/hash/cc4af25fa9d2d5c953496579b75f6f6c-Abstract.html), [2](https://arxiv.org/pdf/2310.07220), [3](https://arxiv.org/pdf/2112.01195)) that utilize Dreamer as the backbone. It's hard to understand how well proposed method works without direct comparison.

**Questions:**

Could you please clarify your argumentation for choosing entropy of stochastic part of learned state representation as the main exploration objective? It is not intuitively understandable how exploration of states with high-entropy stochastic part benefits exploration. As written in paper (lines 176-182), high-entropy stochastic part is usually present for states with high transition function uncertainty regardless of how well state is already explored (which is understandable as the main purpose of stochastic state part in Dreamer is to represent possible uncertainty in transition function), and it's unclear that such high-entropy stochastic part will be present for not explored states (especially with low uncertainty). That's why entropy-based exploration usually target to maximize entropy over states visited during episode: $H(s | \pi)$ rather than uncertainty in transition function.

---

> ### Author Response · Authors · 2025-11-27
>
> W1: We agree that prior work has explored model-based exploration on top of Dreamer, but these methods are structurally quite different from our setting. Maximum-Entropy Dreamer modifies the training objective to estimate and optimize posterior entropy via an additional value head, whereas our method keeps Dreamer’s loss unchanged and adds a lightweight, inference-time planner. COPLANNER relies on Gaussian variance propagation and full world-model ensembles; this is tightly coupled to a specific uncertainty parametrization and carries substantially higher compute and implementation cost, while our approach only assumes access to an RSSM with a stochastic latent and uses entropy as a model-agnostic signal. “Discovering and Achieving Goals via World Models” augments the transition function with a learned goal predictor, which constitutes a fundamental architectural rewrite of Dreamer rather than a drop-in planner.
>
> Given these substantial differences, a fully controlled, resource-matched comparison would require non-trivial re-engineering and is beyond our current compute budget. Our aim in this paper is instead to show that, holding the world model and training objective fixed, an inference-time entropy-seeking planner can consistently improve over base Dreamer (and anticipatory baselines) in terms of data efficiency. We see integrating such planners with more elaborate model-based exploration objectives as complementary future work.
>
> W2: Thank you for raising this. In Dreamer’s RSSM, the stochastic latent is not intended to capture only aleatoric environment noise, but the model’s remaining predictive uncertainty after training (Hafner et al., 2023). Because the KL term drives the prior to match the posterior, the prior variance (and thus prior entropy) increases exactly at timesteps where the posterior repeatedly applies large corrections. This happens disproportionately in under-explored or poorly modelled regions, since familiar transitions lead to small posterior updates and low-entropy priors. In this sense, high prior entropy is not just “transition noise”; it reflects where the world model expects to incur the largest prediction correction, which empirically correlates with epistemic uncertainty and novelty. To prevent the known failure mode where aleatoric stochasticity inflates entropy even for familiar states, our method conditions the trajectory scoring on both reward and entropy. Thus, instead of selecting from all states the model is informed by both high reward and high entropy where higher reward mitigates the risks of choosing states that only captures aleatoric uncertainty. Within these the trajectory with the highest entropy is selected. As reward is completely uncorrelated to aleatoric noise, the model is constrained by the high reward filter and thus it refrains from overoptimizing towards uncertainty and thus aleatoric uncertainty. Classical state-entropy maximization H(s∣π) is complementary but does not exploit the model’s learned posterior-correction structure.

---

### Official Review · Reviewer_2cKJ · 2025-11-01

**Soundness:** 3
**Presentation:** 3
**Contribution:** 3
**Rating:** 6
**Confidence:** 3

**Summary:**

The paper proposes an inference time planning layer for model based reinforcement learning that exploits the world model beyond training to drive purposeful exploration. Building on DreamerV3, the authors generate short horizon imagined rollouts from the current latent state and select the plan whose latent prior has the highest cumulative entropy, which they argue anticipates informative states rather than rewarding novelty only after the fact. A lightweight reactive hierarchical controller trained with PPO learns when to commit to a chosen plan versus replan, with a shaped meta reward that combines environment return and latent prior entropy over a window. The paper motivates entropy seeking via an information gain view of the Dreamer KL objective, discusses pitfalls from aleatoric uncertainty and hidden epistemic uncertainty, and proposes practical mitigations by conditioning candidate plans on a greedy actor before ranking by entropy. Experiments on MiniWorld mazes, Crafter, and DeepMind Control show faster exploration, improved sample efficiency, and equal or better final performance compared to Dreamer without planning and a Plan2Explore baseline. An ablation that replans every step demonstrates that commitment is important and outperforms myopic MPC style replanning.

**Strengths:**

- Clear conceptual framing that links latent prior entropy to information gain in Dreamer and uses it to steer exploration proactively rather than retroactively.
- Practical, model agnostic planner that can wrap existing Dreamer style agents without retraining the actor or replacing the backbone. The gating policy is simple and the squared threshold trick reduces excessive replanning.
- Balanced objective at the meta level that encourages both task progress and coverage of uncertain latent regions, leading to reasoned rather than random exploration.
- Experiments across three regimes with distinct challenges. Results report five seeds on MiniWorld and DMC and three on Crafter, include a strong model based baseline in Plan2Explore, and show consistent gains in sample efficiency. The ablation demonstrates that commitment matters compared to myopic MPC style replanning.
- Implementation details such as inputs to the meta controller, sequence length for meta advantages, and seeds are documented, and a reproducibility statement is provided.

**Weaknesses:**

- The information gain argument motivates seeking high prior entropy, but the claimed min max coupling of world model learning and exploration is more of an interpretation than a concrete change to the Dreamer objective. The method modifies data collection and planning, not the training loss, and this distinction should be made explicit.
- Prior entropy conflates epistemic and aleatoric uncertainty. The paper mitigates this by conditioning on greedy actor proposals and considering reward, yet a direct comparison to ensemble disagreement or epistemic proxies at equal compute would strengthen the case.
- Compute overhead at inference time is nontrivial due to generating 64 imagined rollouts per step and training a PPO meta policy. The wall clock budget is matched for Crafter and DMC, but there is limited profiling of per step latency and throughput relative to Dreamer alone.
- Omission of tuned model free pixel baselines is understandable, yet it leaves open how the proposed planner compares when the best DrQ or SAC variants are properly tuned under the same wall clock and seed budgets.
- Some design choices appear sensitive and are not fully explored, such as the rollout horizon, the number of candidates, and the weight implicit in the meta reward that balances return and entropy. A small sweep or sensitivity analysis would increase confidence in robustness.
- The conclusion mentions instabilities from inflating the KL objective and recommends reinforcing the model, but the main text does not detail the exact KL weighting schedule or regularization that was used. This missing detail hinders reproducibility.

**Questions:**

- How sensitive are results to the candidate count N and the imagined horizon H used for entropy accumulation, both in terms of performance and wall clock cost per environment step?
- Can you clarify whether any explicit change in the Dreamer KL coefficient or loss weighting was used in practice? The conclusion mentions inflating the KL objective. Please specify schedules and values if applicable.
- How does the method compare to Plan2Explore when matching compute at a finer granularity, for example the same number of imagined rollouts per step and the same world model updates? If ensembles are permitted for Plan2Explore under the same wall clock, do your gains persist?
- How robust is the method to model bias early in training when the world model is inaccurate? For example, does the entropy signal overprioritize parts of the latent space that are falsely uncertain due to poor reconstructions?

---

> ### Author Response · Authors · 2025-11-27
>
> We thank the reviewer for the detailed feedback. We have revised the manuscript to clarify the method’s scope, add timing, and include sensitivity analysis. Below we respond point-by-point and indicate the relevant changes in the paper.
>
> W1: Min–max coupling is now stated as interpretation only in contributions, we explicitly say we do not modify dreamer objective in line 80.
>
> W2: Plan2explore largely uses the same amount of compute (on the scale of base dreamer) and we have already included this for an ensemble disagreement baseline, we agree that further comparisons would strengthen the case but the experimental suite is quite resource and time intensive, we instead wanted to show how changing the data collection technique improves upon base dreamer and we have illustrated that in this paper.
>
> W3: We now provide a timing analysis as requested in Sec 5.5 with full profiling tables in Appendix F. We find in practice that the time is not that affected by plan choices as the gpu processes all rollouts in one timestep in one batch, and it scales linearly with timestep.
>
> W4: This is indeed an interesting avenue to pursue future work in, but as underlying models become more complex and associated planners become more intelligent, they will require much more compute and wall clock budget. This is not a method that increases wall clock efficiency but instead is a method that increases data efficiency. When compared to model free baselines, _if_ the model free baseline converges it will likely be much quicker (less wall clock time) and more efficient (less raw compute).
>
> W5: We now provide a sensitivity analysis as requested in Section 5.4 and Appendix G.1 with respect to the number of candidates (N=256, 64, 16), meta reward weight (w=0.0, 0.5, 1.0), PPO advantage sequence length (SL = 32, 8, 2). We do not sweep the rollout horizon as greater than 15 steps is not recommended (the dreamer model is trained with a rollout horizon of 15 by default). Reducing it will not hurt performance for a short while as it is likely that dreamer only makes practical rollouts for a fraction of it’s allocated steps (Translating Latent State World Model Plans into Natural Language – Barker & Leonetti), but when the rollout horizon is ablated beyond Dreamer’s real plan horizon, the performance will start to suffer. This test will only convey base DreamerV3’s realistic plan horizon in each of these environments rather than anything about the planner.
>
> W6: We use a constant free-bits KL objective throughout training: dyn_scale = 0.5, rep_scale = 0.1, and kl_free = 1.0. In each update we feed those fixed values into RSSM.kl_loss, which clips both the dynamics and representation KL terms at the 1.0-nat free-bits floor before reweighting them. There is no annealing or other scheduling, these multipliers and the free-bits regularizer stay constant for the entire run. It should be noted that we do have not modified this from the base dreamer v3 approach.
>
> Q1: Covered in the timing analysis and sensitivity sweep – Sections 5.4, 5.5, Appendix G. Answer – overall wall clock cost increased by about 15%~20% (about 30 minutes accrued per 100,000 steps), performance (speed to convergence) increased by a similar amount. Low sensitivity across N.
>
> Q2: No, the KL objective is said to be inflated because the dataset is filled with samples that have high entropy, and we can see a higher NaN / error rate as a result when we enable our method in crafter (it is stable in dmc and miniworld). While this is because of the environment, it also indicates that the actor is being steered towards states where the game engine fails, possibly because of the game engine having an undefined reward or an undefined transition, reflecting effective exploration.
>
> Q3: Yes, P2E and our method roughly take up similar amounts of time. However, P2E is less data efficient than both our method and base dreamer, as shown in their paper – they reach convergence later than base dreamer does.
>
> Q4: At first, when the encoder is learning all transitions will have high entropy and unknown reward so it is as if trajectories are being randomly selected.
>
> Afterward, when the transition model has not been trained sufficiently, we find that this method is biased by the transition model but that this helps the transition model as poor reconstructions imply that less data representing these transitions have been collected.

---

### Author Response · Authors · 2025-12-03
**Summary of Revisions and Response to Reviews**

Thank you for overseeing our submission, and many thanks to the reviewers for the careful and constructive feedback. We have taken their comments seriously, made targeted revisions, and believe the paper has improved substantially as a result.

Our paper proposes a lightweight, inference-time planner for Dreamer-style world-model agents that steers exploration towards trajectories with high latent prior entropy. The planner scores short imagined trajectories by a shaped meta-objective combining task reward and prior entropy, and a PPO-based meta-controller learns when to commit to a plan versus generating a new one. This leaves the Dreamer-style MBRL backbone and loss unchanged, works with both continuous and discrete RSSMs, and can be added as a drop-in wrapper. Empirically, we observe consistent gains in sample efficiency on MiniWorld mazes, DMC-Vision, and Crafter, often reaching the same performance in roughly half the environment steps as base Dreamer. We also provide comparisons to Plan2Explore as an anticipatory exploration baseline.

**High-level paper changes are addressed below.**

**Introduction & motivations.**
The introduction now explicitly motivates an anticipatory entropy-seeking planner for world-model agents and clearly states that this is meant to complement, not replace, retrospective intrinsic-reward methods. We emphasise in the introduction and Sec.~4 that we do not modify Dreamer’s training objective; our contribution is a purely inference-time planning layer.

**Method.**
Sections 3–4 and Appendix B have been rewritten for clarity without changing the underlying algorithm. We now introduce the MDP/POMDP formalism and its connection to Dreamer’s RSSM, and derive the information-gain view of Dreamer’s KL term while presenting the “min–max” coupling explicitly as an interpretation and extension, not a new loss. Section 4.1 shows how prior entropy can be seen as a proxy for anticipated posterior correction and how it contrasts and complements the KL loss inherent to many MBRL methods. We also clarify that we use entropy because it is well-defined for both Gaussian and categorical latents, while being highly relevant toward maximising information gain.

**Experiments.**
We increased Crafter from 3 to 5 seeds, and we now report learning curves with 95% confidence intervals (±2×SEM). Legends and naming have been cleaned up (Dreamer vs. Dreamer+Planner vs. Plan2Explore vs. PPO). We have correspondingly made our claims more precise and better grounded: we emphasise consistent sample-efficiency improvements and support them with the reported confidence intervals.

**Ablations and timing.**
We expand the ablation section and appendices to report sensitivity to candidate count $N$ and meta-horizon $L$ on four DMC tasks, together with a justification for keeping the rollout horizon aligned with Dreamer’s effective planning range. We further ablate the meta-reward composition (entropy-only, reward-only, 50/50 mix) on DMC and Crafter, showing that performance degrades gracefully as $N$ is reduced, that each task’s required attention span is reflected in the choice of $L$, and that both reward- and entropy-based planning remain robust. Finally, we add a timing analysis section with tables in Appendix F, reporting per-step planning latency, scaling with horizon and candidate count, and PPO meta-policy overhead.

**Reviewer-specific concerns are addressed below.**

**Reviewer 2cKJ.** We explicitly frame the KL/IG coupling as interpretive only; add timing and sensitivity analyses; document KL settings and confirm that we keep the Dreamer loss unchanged; and discuss compute trade-offs versus Plan2Explore and model-free baselines.

**Reviewer WBUJ.** We expand the explanation of why prior entropy of the stochastic latent correlates with anticipated model correction.

**Reviewer KowQ.** We overhaul Sections 3–4 and Appendix B to make the formalism, equations, and the role of the meta-planner precise and self-contained, and we strengthen the statistical presentation and discussion of effect sizes across DMC and Crafter.

**Reviewer WShA.** We clarify the limits of entropy as an epistemic proxy and why we nevertheless choose it for model-agnosticity (continuous and discrete latents). We also adopt their suggestions on confidence intervals and figure legends, and make explicit that our anticipatory planner is orthogonal to retrospective intrinsic-reward methods rather than in opposition to them.

Overall, we believe the manuscript is now significantly clearer, more careful in its claims, and more thoroughly evaluated, while keeping the core idea unchanged: a lightweight, inference-time entropy-seeking planner that improves the *sample efficiency* of MBRL agents without modifying their training losses. We hope this addresses the reviewers’ concerns and that you will consider the revised version favourably.

---

### Meta-Review · Area_Chair_87cF · 2025-12-29

**Summary:**

The main three concerns seemed to be presentation [2cKJ, KowQ], baselines [2cKJ, WBUJ, WShA], the distinction between aleatoric/epistemic uncertainty [2cKJ, KowQ, WShA], and statistical significance of the results [KowQ, WShA]. The authors made changes to the paper to aid presentation, though reviewers felt that additional changes were needed after the first round of revisions [KowQ, WShA]. The authors did not include the additional baselines requested by the reviewers, arguing that the current baselines (e.g., Plan2Explore) were sufficient to answer the hypotheses. For notions of uncertainty, the paper was revised to clarify that the paper is only using a "proxy" measure of epistemic uncertainty. Additional random seeds were added, but no p-values were provided.

Overall, reviewers were split on this paper, with two voting to accept and two voting to reject. My recommendation is that the paper be rejected for two reasons. First, the gains relative to prior work are not clearly demonstrated (no p-values, overlapping confidence intervals). Second, the paper is missing some contextualization of the approach relative to prior methods. All four reviewers came away with some level of confusion about how the method relates to alternative methods builds from similar components. While the actual _method_ does indeed seem distinct, the paper's _writing_ seems to require revisions to help to explain the "lay of the land" to readers.

**Reviewer Concerns:**

(see below)

**Reviewer Scores:**

2cKJ: 6 --> 6
* [+] The method modifies data collection and planning, not the training loss, and this distinction should be made explicit: Authors have revised this part of the paper
* [+] direct comparison to ensemble disagreement or epistemic proxies at equal compute would strengthen the case: authors agreed, noting that the plan2explore baseline already provides an ensemble disagreement baseline
* [+] profiling of per step latency and throughput relative to Dreamer alone: Authors added this in appendix F
* [-] how the proposed planner compares when the best DrQ or SAC variants are properly tuned under the same wall clock and seed budgets: authors argued that the proposed method improves data effiency, not wall clock efficiency. I think this response would be stronger if it had actually provided data comparing with DrQ or SAC.
* [+] A small sweep or sensitivity analysis would increase confidence in robustness: Authors provided this ablation in Appendix G.1.
* [+] Text does not detail the exact KL weighting schedule: Authors provided these details.

WBUJ: 4 --> 6
* [+] how does the proposed method compare with alternative model-based exploration techniques that are also based upon the Dreamer backbone? Authors argue that these alternative methods are "structurally quite different," noting that the proposed method doesn't require changing the world model or training objective. If I (AC) understand correctly, the proposed method only changes steps that happen at inference-time, whereas these alternative methods change steps that happen during training-time.

KowQ: 2 --> 2
* [/] incomplete or missing formalism, description and intuition: Authors revised the paper to include more notation and description in the preliminaries. I (AC) think that some notation is still a bit confusing (e.g., what is the difference between $p_\phi$ and $\hat{p}_\phi$? where is $p_C$ defined?)
* [/] insufficient distinction between aleatoric and epistemic uncertainty: authors clarified that they "treat prior entropy as a proxy for epistemic uncertainty."
* [/] So why do the authors expect the prior entropy to be a good exploration signal? Authors argued that they expect entropy to be highest "precisely where the model expects strong posterior corrections." Both this response and the one prior might be strengthened with numerical evidence.
* [/] not a sufficient level of evidence to claim that the proposed method has any systematic effect (no clearly separated standard deviations): authors noted that the "improvements are not always statistically separated at the 95% level" and softened the wording in the paper. I agree with the reviewer that, just looking at Figures 4/5, it's hard to tell whether the results are statistically significant, and the paper could be strengthened by providing p-values.

WShA: 2 --> 2
* [-] unclear if the method makes sense in stochastic environments: authors acknowledged this limitation, but note that planning aims to maximize both entropy and reward
* [-] in deterministic environments, why would we prefer entropy over variance as a metric for novelty? Authors argue that only entropy is applicable to both discrete and continuous RVs. I (AC) disagree with this argument --- one can easily compute the variance of discrete RVs.
* [/] why would entropy of the transition model be a better metric for novelty that ensemble disagreement? Authors noted that they empirically compared with an ensemble disagreement method (Plan2Explore) and find it performs worse. However, the explanation for why ("frequent planning without commitment") isn't clearly substantiated with numerical evidence.
* [+] should the intrinsic reward be included in the value function? I think the authors are arguing that the intrinsic rewards are summed across multiple steps of the model rollout, thereby making the planning less myopic (than greedily maximizing the intrinsic reward at the next time step).
* [-] number of random seeds, statistical significance of the results: Authors have added additional random seeds for 5 total in each plot and plotted the 95% confidence interval. No p-values are provided.
* [-] comparison with Epistemic Monte Carlo Tree Search: Authors argue that it would be challenging to compare to this baseline because this prior method "targets low-dimensional, discrete-state domains"
In the rebuttal, the reviewer responded noting that many of these concerns had not been addressed

---

### Decision · Program_Chairs · 2026-01-26

Reject